# INFINITYGAN: TOWARDS INFINITE-PIXEL IMAGE SYNTHESIS

**Chieh Hubert Lin**[1] **Hsin-Ying Lee**[2] **Yen-Chi Cheng**[3] **Sergey Tulyakov**[2] **Ming-Hsuan Yang**[1,4,5]
[1]UC Merced  [2]Snap Inc.  [3]Carnegie Mellon University  [4]Yonsei University  [5]Google Research

## ABSTRACT

We present a novel framework, InfinityGAN, for arbitrary-sized image generation. The task is associated with several key challenges. First, scaling existing models to an arbitrarily large image size is resource-constrained, in terms of both computation and availability of large-field-of-view training data. InfinityGAN trains and infers in a seamless patch-by-patch manner with low computational resources. Second, large images should be locally and globally consistent, avoid repetitive patterns, and look realistic. To address these, InfinityGAN disentangles global appearances, local structures, and textures. With this formulation, we can generate images with spatial size and level of details not attainable before. Experimental evaluation validates that InfinityGAN generates images with superior realism compared to baselines and features parallelizable inference. Finally, we show several applications unlocked by our approach, such as spatial style fusion, multimodal outpainting, and image inbetweening. All applications can be operated with arbitrary input and output sizes.

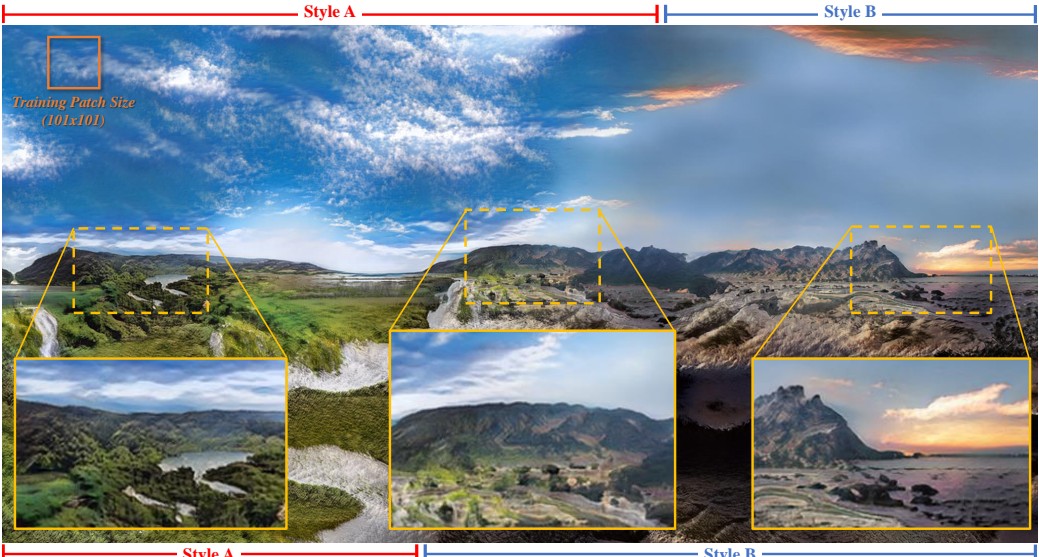

Figure 1: **Synthesizing infinite-pixel images from finite-sized training data.** A $1024 \times 2048$ image composed of 242 patches, independently synthesized by InfinityGAN with spatial fusion of two styles. The generator is trained on $101 \times 101$ patches (e.g., marked in top-left) sampled from $197 \times 197$ real images. Note that training and inference (of any size) are performed on a single GTX TITAN X GPU. Zoom-in for better experience.

## 1 INTRODUCTION

> *"To infinity and beyond!"*             *– Buzz Lightyear*

Generative models witness substantial improvements in resolution and level of details. Most improvements come at a price of increased training time (Gulrajani et al., 2017; Mescheder et al., 2018), larger model size (Balaji et al., 2021), and stricter data requirements (Karras et al., 2018). The most recent works synthesize images at $1024 \times 1024$ resolution featuring a high level of details

---

All codes, datasets, and trained models are publicly available. Project page: https://hubert0527.github.io/infinityGAN/

and fidelity. However, models generating high resolution images usually still synthesize images of limited field-of-view bounded by the training data. It is not straightforward to scale these models to generate images of arbitrarily large field-of-view. Synthesizing infinite-pixel images is constrained by the finite nature of resources. Finite computational resources (e.g., memory and training time) set bounds for input receptive field and output size. A further limitation is that there exists no infinite-pixel image dataset. Thus, to generate infinite-pixel images, a model should learn the implicit global structure without direct supervision and under limited computational resources.

Repetitive texture synthesis methods (Efros & Leung, 1999; Xian et al., 2018) generalize to large spatial sizes. Yet, such methods are not able to synthesize real-world images. Recent works, such as SinGAN (Shaham et al., 2019) and InGAN (Shocher et al., 2019), learn an internal patch distribution for image synthesis. Although these models can generate images with arbitrary shapes, in Section 4.1, we show that they do not infer structural relationships well, and fail to construct plausible holistic views with spatially extended latent space. A different approach, COCO-GAN (Lin et al., 2019), learns a coordinate-conditioned patch distribution for image synthesis. As shown in Figure 4, despite the ability to slightly extend images beyond the learned boundary, it fails to maintain the global coherence of the generated images when scaling to a $2\times$ larger generation size.

*How to generate infinite-pixel images?* Humans are able to guess the whole scene given a partial observation of it. In a similar fashion, we aim to build a generator that trains with image patches, and inference images of unbounded arbitrary-large size. An example of a synthesized scene containing globally-plausible structure and heterogeneous textures is shown in Figure 1.

We propose InfinityGAN, a method that trains on a finite-pixel dataset, while generating infinite-pixel images at inference time. InfinityGAN consists of a neural implicit function, termed *structure synthesizer*, and a padding-free StyleGAN2 generator, dubbed *texture synthesizer*. Given a global appearance of an infinite-pixel image, the structure synthesizer samples a sub-region using coordinates and synthesizes an intermediate local structural representations. The texture synthesizer then seamlessly synthesizes the final image by parts after filling the fine local textures to the local structural representations. InfinityGAN can infer a compelling global composition of a scene with realistic local details. Trained on small patches, InfinityGAN achieves high-quality, seamless and arbitrarily-sized outputs with low computational resources—a single TITAN X to train and test.

We conduct extensive experiments to validate the proposed method. Qualitatively, we present the everlastingly long landscape images. Quantitatively, we evaluate InfinityGAN and related methods using user study and a proposed ScaleInv FID metric. Furthermore, we demonstrate the efficiency and efficacy of the proposed methods with several applications. First, we demonstrate the flexibility and controllability of the proposed method by spatially fusing structures and textures from different distributions within an image. Second, we show that our model is an effective deep image prior for the image outpainting task with the image inversion technique and achieves multi-modal outpainting of arbitrary length from arbitrarily-shaped inputs. Third, with the proposed model we can divide-and-conquer the full image generation into independent patch generation and achieve $7.2\times$ of inference speed-up with parallel computing, which is critical for high-resolution image synthesis.

## 2 RELATED WORK

**Latent generative models.** Existing generative models are mostly designed to synthesize images of fixed sizes. A few methods (Karras et al., 2018; 2020) have been recently developed to train latent generative models on high-resolution images, up to $1024\times1024$ pixels. However, latent generative models generate images from dense latent vectors that require synthesizing all structural contents at once. Bounded by computational resources and limited by the learning framework and architecture, these approaches synthesize images of certain sizes and are non-trivial to generalize to different output size. In contrast, patch-based GANs trained on image patches (Lin et al., 2019; Shaham et al., 2019; Shocher et al., 2019) are less constrained by the resource bottleneck with the synthesis-by-part approach. However, (Shaham et al., 2019; Shocher et al., 2019) can only model and repeat internal statistics of a single image, and (Lin et al., 2019) can barely extrapolate few patches beyond the training size. ALIS (Skorokhodov et al., 2021) is a concurrent work that also explores synthesizing infinite-pixel images. It recursively inbetweens latent variable pairs in the horizontal direction. We further discuss the method in Appendix A.1. Finally, autoregressive models (Oord et al., 2016; Razavi et al., 2019; Esser et al., 2021) can theoretically synthesize at arbitrary image sizes. Despite (Razavi et al., 2019) and (Esser et al., 2021) showing unconditional images synthesis at $1024\times1024$ resolution, their application in infinite-pixel image synthesis has not yet been well-explored.

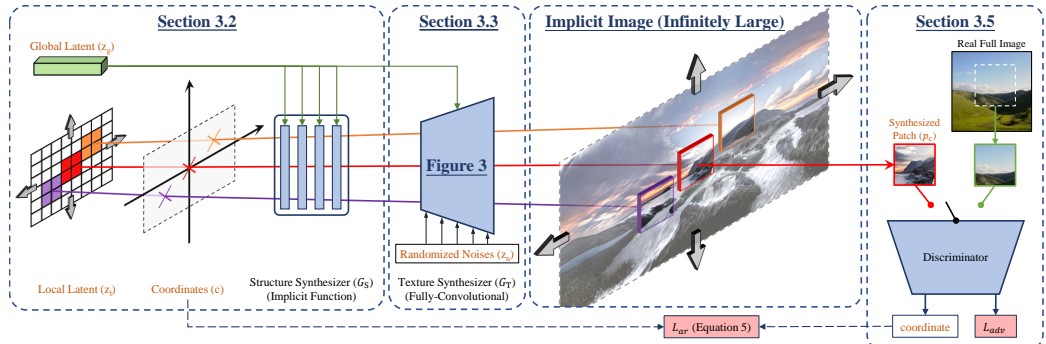

Figure 2: **Overview.** The generator of InfinityGAN consists of two modules, a structure synthesizer based on a neural implicit function, and a fully-convolutional texture synthesizer with all positional information removed (see Figure 3). The two networks take four sets of inputs, a global latent variable that defines the holistic appearance of the image, a local latent variable that represents the local and structural variation, a continuous coordinate for learning the neural implicit structure synthesizer, and a set of randomized noises to model fine-grained texture. InfinityGAN synthesizes images of arbitrary size by learning spatially extensible representations.

**Conditional generative models.** Numerous tasks such as image super-resolution, semantic image synthesis, and image extrapolation often showcase results over $1024 \times 1024$ pixels. These tasks are less related to our setting, as most structural information is already provided in the conditional inputs. We illustrate and compare the characteristics of these tasks against ours in Appendix B.

**Image outpainting.** Image outpainting (Abdal et al., 2020; Liu et al., 2021; Sabini & Rusak, 2018; Yang et al., 2019) is related to image inpainting (Liu et al., 2018a; Yu et al., 2019) and shares similar issues that the generator tends to copy-and-paraphrase the conditional input or create mottled textural samples, leading to repetitive results especially when the outpainted region is large. InOut (Cheng et al., 2021) proposes to outpaint image with GANs inversion and yield results with higher diversity. We show that with InfinityGAN as the deep image prior along with InOut (Cheng et al., 2021), we obtain the state-of-the-art outpainting results and avoids the need of sequential outpainting. Then, we demonstrate applications in arbitrary-distant image inbetweening, which is at the intersection of image inpainting (Liu et al., 2018a; Nazeri et al., 2019; Yu et al., 2019) and outpainting research.

**Neural implicit representation.** Neural implicit functions (Park et al., 2019; Mescheder et al., 2019; Mildenhall et al., 2020) have been applied to model the structural information of 3D and continuous representations. Adopting neural implicit modeling, our query-by-coordinate synthesizer is able to model structural information effectively. Some recent works (DeVries et al., 2021; Niemeyer & Geiger, 2021; Chan et al., 2021) also attempt to integrate neural implicit function into generative models, but aiming at 3D-structure modeling instead of extending the synthesis field-of-view.

## 3 PROPOSED METHOD

### 3.1 OVERVIEW

An arbitrarily large image can be described globally and locally. Globally, images should be coherent and hence global characteristics should be expressible by a compact *holistic appearance* (e.g., a medieval landscape, ocean view panorama). Therefore, we adopt a fixed holistic appearance for each infinite-pixel image to represent the high-level composition and content of the scene. Locally, a close-up view of an image is defined by its local structure and texture. The structure represents objects, shapes and their arrangement within a local region. Once the structure is defined, there exist multiple feasible appearances or textures to render realistic scenes. At the same time, structure and texture should conform to the global *holistic appearance* to maintain the visual consistency among the neighboring patches. Given these assumptions, we can generate an infinite-pixel image by first sampling a global holistic appearance, then spatially extending local structures and textures following the holistic appearance.

Accordingly, the InfinityGAN generator $G$ consists of a *structure synthesizer* $G_S$ and a *texture synthesizer* $G_T$. $G_S$ is an implicit function that samples a sub-region with coordinates and creates local structural features. $G_T$ is a fully convolutional StyleGAN2 (Karras et al., 2020) modeling textural properties for local patches and rendering final image. Both modules follow a consistent holistic appearance throughout the process. Figure 2 presents the overview of our framework.

## 3.2 STRUCTURE SYNTHESIZER ($G_S$)

Structure synthesizer is a neural implicit function driven by three sets of latent variables: A global latent vector $\mathbf{z}_g$ representing the holistic appearance of the infinite-pixel image (also called *implicit image* since the whole image is never explicitly sampled), a local latent tensor $\mathbf{z}_l$ expressing the local structural variation of the image content, and a coordinate grid $\mathbf{c}$ specifying the location of the patches to sample from the implicit image. The synthesis process is formulated as:

$$\mathbf{z}_S = G_S(\mathbf{z}_g, \mathbf{z}_l, \mathbf{c}), \tag{1}$$

where $\mathbf{z}_S$ denotes the structural latent variable that is later used as an input to the texture synthesizer.

We sample $\mathbf{z}_g \in \mathbb{R}^{D_{\mathbf{z}_g}}$ from a unit Gaussian distribution once and inject $\mathbf{z}_g$ into every layer and pixel in $G_S$ via feature modulation (Huang & Belongie, 2017; Karras et al., 2020). As local variations are independent across the spatial dimension, we independently sample them from a unit Gaussian prior for each spatial position of $\mathbf{z}_l \in \mathbb{R}^{H \times W \times D_{\mathbf{z}_l}}$, where $H$ and $W$ can be arbitrarily extended.

We then use coordinate grid $\mathbf{c}$ to specify the location of the target patches to be sampled. To be able to condition $G_S$ with coordinates infinitely far from the origin, we introduce a prior by exploiting the nature of landscape images: (a) self-similarity for the horizontal direction, and (b) rapid saturation (e.g., land, sky or ocean) for the vertical direction. To implement this, we use the positional encoding for the horizontal axis similar to (Vaswani et al., 2017; Tancik et al., 2020; Sitzmann et al., 2020). We use both sine and cosine functions to encode each coordinate for numerical stability. For the vertical axis, to represent saturation, we apply the tanh function. Formally, given horizontal and vertical indexes $(i_x, i_y)$ of $\mathbf{z}_l$ tensor, we encode them as $\mathbf{c} = (\tanh(i_y), \cos(i_x/T), \sin(i_x/T))$, where $T$ is the period of the sine function and $\mathbf{c}$ controls the location of the patch to generate.

To prevent the model from ignoring the variation of $\mathbf{z}_l$ and generating repetitive content by following the periodically repeating coordinates, we adopt a mode-seeking diversity loss (Mao et al., 2019; Lee et al., 2020) between a pair of local latent variables $\mathbf{z}_{l_1}$ and $\mathbf{z}_{l_2}$ while sharing the same $\mathbf{z}_g$ and $\mathbf{c}$:

$$\mathcal{L}_{\text{div}} = \|\mathbf{z}_{l_1} - \mathbf{z}_{l_2}\|_1 \ / \ \|G_S(\mathbf{z}_g, \mathbf{z}_{l_1}, \mathbf{c}) - G_S(\mathbf{z}_g, \mathbf{z}_{l_2}, \mathbf{c})\|_1. \tag{2}$$

Conventional neural implicit functions produce outputs for each input query independently, which is a pixel in $\mathbf{z}_l$ for InfinityGAN. Such a design causes training instabilities and slows convergence, as we show in Figure 37. We therefore adopt the feature unfolding technique (Chen et al., 2021) to enable $G_S$ to account for the information in a broader neighboring region of $\mathbf{z}_l$ and $\mathbf{c}$, introducing a larger receptive field. For each layer in $G_S$, before feeding forward to the next layer, we apply a $k \times k$ feature unfolding transformation at each location $(i, j)$ of the origin input $f$ to obtain the unfolded input $f'$:

$$f'_{(i,j)} = \text{Concat}(\{f(i + n, j + m)\}_{n,m \in \{-k/2, k/2\}}), \tag{3}$$

where Concat($\cdot$) concatenates the unfolded vectors in the channel dimension. In practice, as the grid-shaped $\mathbf{z}_l$ and $\mathbf{c}$ are sampled with equal spacing between consecutive pixels, the feature unfolding can be efficiently implemented with CoordConv (Liu et al., 2018b).

## 3.3 TEXTURE SYNTHESIZER ($G_T$)

Texture synthesizer aims to model various realizations of local texture given the local structure $\mathbf{z}_S$ generated by the structure synthesizer. In addition to the holistic appearance $\mathbf{z}_g$ and the local structural latent $\mathbf{z}_S$, texture synthesizer uses noise vectors $\mathbf{z}_n$ to model the finest-grained textural variations that are difficult to capture by other variables. The generation process can be written as:

$$\mathbf{p}_c = G_T(\mathbf{z}_S, \mathbf{z}_g, \mathbf{z}_n), \tag{4}$$

where $\mathbf{p}_c$ is a generated patch at location $\mathbf{c}$ (i.e., the $\mathbf{c}$ used in Eq 1 for generating $\mathbf{z}_S$).

We implement upon StyleGAN2 (Karras et al., 2020). First, we replace the fixed constant input with the generated structure $\mathbf{z}_S$. Similar to StyleGAN2, randomized noises $\mathbf{z}_n$ are added to all layers of $G_T$, representing the local variations of fine-grained textures. Then, a mapping layer projects $\mathbf{z}_g$ to a style vector, and the style is injected into all pixels in each layer via feature modulation. Finally, we remove all zero-paddings from the generator, as shown in Figure 3(b).

Both zero-padding and $G_S$ can provide positional information to the generator, and we later show that positional information is important for generator learning in Section 4.2. However, it is necessary to remove all zero-paddings from $G_T$ for three major reasons. First, zero-padding has a consistent pattern during training, due to the fixed training image size. Such a behavior misleads

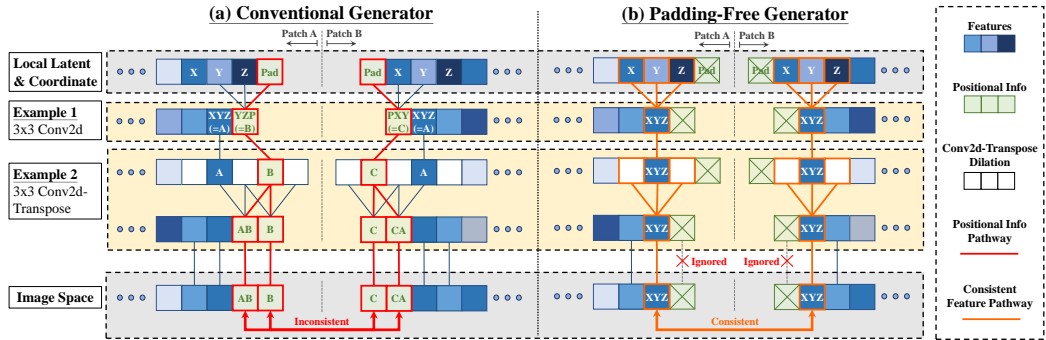

Figure 3: **Padding-free generator.** (Left) Conventional generators synthesize inconsistent pixels due to the zero-paddings. Note that the inconsistency region grows exponentially as the network deepened. (Right) In contrast, our padding-free generator can synthesize consistent pixel value regardless of the position in the model receptive field. Such a property facilitates spatially-independently generating patches and forming into a seamless image with consistent feature values.

the generator to memorize the padding pattern, and becomes vulnerable to unseen padding patterns while attempting to synthesize at a different image size. The third column of Figure 4 shows when we extend the input latent variable of the StyleGAN2 generator multiple times, the center part of the features does not receive expected coordinate information from the paddings, resulting in extensively repetitive textures in the center area of the output image. Second, zero-paddings can only provide positional information within a limited distance from the image border. However, while generating infinite-pixel images, the image border is considered infinitely far from the generated patch. Finally, as shown in Figure 3, the existence of paddings hampers $G_\mathrm{T}$ from generating separate patches that can be composed together. Therefore, we remove all paddings from $G_\mathrm{T}$, facilitating the synthesis-by-parts of arbitrary-sized images. We refer to the proposed $G_\mathrm{T}$ as a padding-free generator (PFG).

## 3.4 Spatially Independent Generation

InfinityGAN enables spatially independent generation thanks to two characteristics of the proposed modules. First, $G_\mathrm{S}$, as a neural implicit function, naturally supports independent inference at each spatial location. Second, $G_\mathrm{T}$, as a fully convolutional generator with all paddings removed, can synthesize consistent pixel values at the same spatial location in the implicit image, regardless of different querying coordinates, as shown in Figure 3(b). With these properties, we can independently query and synthesize a patch from the implicit image, seamlessly combine multiple patches into an arbitrarily large image, and maintain *constant* memory usage while synthesizing images of any size.

In practice, having a single center pixel in a $\mathbf{z}_\mathrm{S}$ slice that aligns to the center pixel of the corresponding output image patch can facilitate $\mathbf{z}_\mathrm{l}$ and $\mathbf{c}$ indexing. We achieve the goal by shrinking the StyleGAN2 blur kernel size from 4 to 3, causing the model to generate odd-sized features in all layers, due to the convolutional transpose layers.

## 3.5 Model Training

The discriminator $D$ of InfinityGAN is similar to the one in the StyleGAN2 method. The detailed architectures of $G$ and $D$ are presented in Appendix D. The two networks are trained with the non-saturating logistic loss $\mathcal{L}_\mathrm{adv}$ (Goodfellow et al., 2014), $R_1$ regularization $\mathcal{L}_{\mathrm{R}_1}$ (Mescheder et al., 2018) and path length regularization $\mathcal{L}_\mathrm{path}$ (Karras et al., 2020). Furthermore, to encourage the generator to follow the conditional distribution in the vertical direction, we train $G$ and $D$ with an auxiliary task (Odena et al., 2017) predicting the vertical position of the patch:

$$\mathcal{L}_\mathrm{ar} = \|\hat{\mathbf{c}}_y - \bar{\mathbf{c}}_y\|_1 \,, \tag{5}$$

where $\hat{\mathbf{c}}_y$ is the vertical coordinate predicted by $D$, and $\bar{\mathbf{c}}_y$ is either (for generated images) $\mathbf{c}_y = \tanh(i_y)$ or (for real images) the vertical position of the patch in the full image. We formulate $\mathcal{L}_\mathrm{ar}$ as a regression task. The overall loss function for the InfinityGAN is:

$$\begin{aligned} \min_D \ & \mathcal{L}_\mathrm{adv} + \lambda_\mathrm{ar}\mathcal{L}_\mathrm{ar} + \lambda_{\mathrm{R}_1}\mathcal{L}_{\mathrm{R}_1} \,, \\ \min_G \ & -\mathcal{L}_\mathrm{adv} + \lambda_\mathrm{ar}\mathcal{L}_\mathrm{ar} + \lambda_\mathrm{div}\mathcal{L}_\mathrm{div} + \lambda_\mathrm{path}\mathcal{L}_\mathrm{path} \,, \end{aligned} \tag{6}$$

where $\lambda$'s are the weights.

Table 1: **Quantitative evaluation on Flickr-Landscape.** Despite we use a disadvantageous setting for our InfinityGAN (discussed in Section 4.1), it still outperforms all baselines after extending the size to 4× larger. Furthermore, the user study shows an over 90% preference favors our Infinity-GAN results. The preference is marked as $x\%$ when $x\%$ of selections prefer the results from the corresponding method over InfinityGAN. [†]The images are first resized to 128 before resizing to 197.

| Method | Image Size | | | FID | ScaleInv FID | | | | Preference v.s. Ours | | Inference |
| --- | --- | --- | --- | --- | --- | --- | --- | --- | --- | --- | --- |
| | Full | Train | Test 8× | Train | 1× | 2× | 4× | 8× | 4× | 8× | Memory |
| SinGAN | 128 | 128 | 1024 | 4.21 | 4.21 | 57.10 | 145.12 | 210.22 | 0.80% | 1.60% | $\mathcal{O}(\text{size}^2)$ |
| COCO-GAN | 128 | 32 | 1024 | 17.52 | 41.32 | 258.51 | 376.69 | 387.15 | 0% | 0% | $\mathcal{O}(1)$ |
| StyleGAN2+NCI | 128 | 128 | 1024 | 4.19 | **4.19** | **18.31** | 79.83 | 189.65 | 9.20% | 7.20% | $\mathcal{O}(\text{size}^2)$ |
| StyleGAN2+NCI (Patched) | 128 | 64 | 1024 | 5.35 | 21.06 | 58.84 | 165.65 | 234.19 | - | - | $\mathcal{O}(\text{size}^2)$ |
| StyleGAN2+NCI+PFG | 197[†] | 101 | 1576 | 86.76 | 90.79 | 126.88 | 211.22 | 272.80 | 0.40% | 1.20% | $\mathcal{O}(1)$ |
| InfinityGAN (Ours) (StyleGAN2+NCI+PFG+$G_\text{S}$) | 197[†] | 101 | 1576 | 11.03 | 21.84 | 28.83 | **61.41** | **121.18** | - | - | $\mathcal{O}(1)$ |

|  | COCO-GAN | SinGAN | StyleGAN+NCI | StyleGAN+NCI+PFG | InfinityGAN (ours) |
| --- | --- | --- | --- | --- | --- |
| Generated Full Image | 128×128 | 128×128 | 128×128 | 197×197 | 197×197 |

Figure 4: **Qualitative comparison.** We show that InfinityGAN can produce more favorable holistic appearances against related methods while testing with an extended size 1024×1024. (NCI: Non-Constant Input, PFG: Padding-Free Generator). More results are shown in Appendix E.

# 4 EXPERIMENTAL RESULTS

**Datasets.** We evaluate the ability of synthesizing at extended image sizes on the Flickr-Landscape dataset consists of 450,000 high-quality landscape images, which are crawled from the Landscape group on Flickr. For the image outpainting experiments, we evaluate with other baseline methods on scenery-related subsets from the Place365 (Zhou et al., 2017) (62,500 images) and Flickr-Scenery (Cheng et al., 2021) (54,710 images) datasets. Note that the Flickr-Scenery here is different from our Flickr-Landscape. For image outpainting task, we split the data into 80%, 10%, 10% for training, validation, and test. All quantitative and qualitative evaluations are conducted on test set.

**Hyperparameters.** We use $\lambda_{ar} = 1$, $\lambda_{div} = 1$, $\lambda_{R_1} = 10$, and $\lambda_{path} = 2$ for all datasets. All models are trained with 101×101 patches cropped from 197×197 real images. Since our InfinityGAN synthesizes odd-sized images, we choose 101 that maintains a sufficient resolution that humans can still recognize its content. On the other hand, 197 is the next output resolution if stacking another upsampling layer to InfinityGAN, which also provides 101×101 patches a sufficient field-of-view. We adopt the Adam (Kingma & Ba, 2015) optimizer with $\beta_1 = 0$, $\beta_2 = 0.99$ and a batch size 16 for 800,000 iterations. More details are presented in Appendix C.

**Metrics.** We first evaluate Fréchet Inception Distance (**FID**) (Heusel et al., 2017) at $G$ training resolution. Then, without access to real images at larger sizes, we assume that the real landscape with a larger FoV will share a certain level of self-similarity with its smaller FoV parts. We accordingly propose a **ScaleInv FID**, which resizes larger images to the training data size with bilinear interpolation, then computes FID. We denote N× ScaleInv FID when the metric is evaluated with images N× larger than the training samples.

**Evaluated Method.** We perform the evaluation on Flickr-Landscape with the following algorithms:

- **SinGAN.** We train an individual SinGAN model for each image. The images at larger sizes are generated by setting spatially enlarged input latent variables. Note that we do not compare with the super-resolution setting from SinGAN since we focus on extending the learned structure rather than super-resolve the high-frequency details.

- **COCO-GAN.** Follow the "Beyond-Boundary Generation" protocol of COCO-GAN, we transfer a trained COCO-GAN model to extended coordinates with a post-training procedure.
- **StyleGAN2 (+NCI).** We replace the constant input of the original StyleGAN2 with a $z_l$ of the same shape, we call such a replacement as "non-constant input (NCI)". This modification enables StyleGAN2 to generate images at different output sizes with different $z_l$ sizes.

## 4.1 GENERATION AT EXTENDED SIZE.

**Additional (unfair) protocols for fairness.** We adopt two additional pre- and post-processing to ensure that InfinityGAN does not take advantage of its different training resolution. To ensure InfinityGAN is trained with the same amount of information as other methods, images are first bilinear interpolated into 128×128 before resized into 197×197. Next, for all testing sizes in Table 4, InfinityGAN generates at 1.54× (=197/128) larger size to ensure final output images share the same FoV with others. In fact, these corrections make the setting disadvantageous for InfinityGAN, as it is trained with patches of 50% FoV, generates 54% larger images for all settings, and requires to composite multiple patches for its 1× ScaleInv FID.

**Quantitative analysis.** For all the FID metrics in Table 1, unfortunately, the numbers are not directly comparable, since InfinityGAN is trained with patches with smaller FoV and at a different resolution. Nevertheless, the trend in ScaleInv FID is informative. It reflects the fact that the global structures generated from the baselines drift far away from the real landscape as the testing FoV enlarges. Meanwhile, InfinityGAN maintains a more steady slope, and surpasses the strongest baseline after 4× ScaleInv FID. Showing that InfinityGAN indeed performs favorably better than all baselines as the testing size increases.

**Qualitative results.** In Figure 4, we show that all baselines fall short of creating reasonable global structures with spatially expanded input latent variables. COCO-GAN is unable to transfer to new coordinates when the extrapolated coordinates are too far away from the training distribution. Both SinGAN and StyleGAN2 implicitly establish image features based on position encoded by zero padding, assuming the training and testing position encoding should be the same. However, when synthesizing at extended image sizes, the inevitable change in the spatial size of the input and the features leads to drastically different position encoding in all model layers. Despite the models can still synthesize reasonable contents near the image border, where the position encoding is still partially correct, they fail to synthesize structurally sound content in the image center. Such a result causes ScaleInv FID to rapidly surge as the extended generation size increases to 1024×1024. Note that at the 16× setting, StyleGAN2 runs out of memory with a batch size of 1 and does not generate any result. In comparison, InfinityGAN achieves reasonable global structures with fine details. Note that the 1024×1024 image from InfinityGAN is created by compositing 121 independently synthesized patches. With the ability of generating consistent pixel values (Section 3.4), the composition is guaranteed to be seamless. We provide more comparisons in Appedix E, a larger set of generated samples in Appendix F, results from models trained at a higher resolution in Appendix G, and a very-long synthesis result in Appendix J.

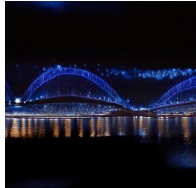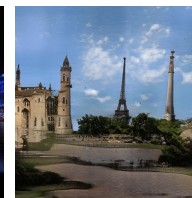

Figure 5: **LSUN bridge and tower.** InfinityGAN synthesize at 512×512 pixels. We provide more details and samples in Appendix H.

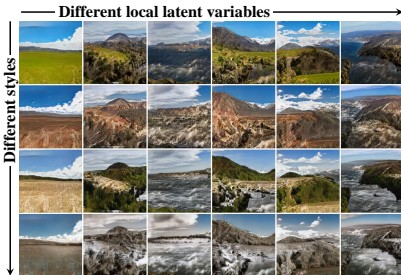

Figure 6: **Diversity.** InfinityGAN synthesizes diverse samples at the same coordinate with different local latent and styles. More samples are shown in Appendix I

In Figure 5, we further conduct experiments on LSUN bridge and tower datasets, demonstrating InfinityGAN is applicable on other datasets. However, since the two datasets are object centric with a low view-angle variation in the vertical direction, InfinityGAN frequently fills the top and bottom area with blank padding textures.

In Figure 6, we switch different $z_l$ and $G_T$ styles (i.e., $z_g$ projected with the mapping layer) while sharing the same **c**. More samples can be found in Appendix I. The results show that the structure

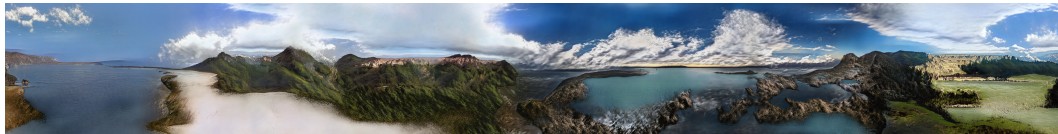

Figure 7: **Spatial style fusion.** We present a mechanism in fusing multiple styles together to increase the interestingness and interactiveness of the generation results. The 512×4096 image fuses four styles across 258 independently generated patches.

Table 2: **Outpainting performance.** The combination of In&Out (Cheng et al., 2021) and InfinityGAN achieves state-of-the-art IS (higher better) and FID (lower better) performance on image outpainting task.

| Method | Place365 | | Flickr-Scenery | |
|---|---|---|---|---|
| | FID ↓ | IS ↑ | FID ↓ | IS ↑ |
| Boundless | 35.02 | 6.15 | 61.98 | 6.98 |
| NS-outpaint | 50.68 | 4.70 | 61.16 | 4.76 |
| In&Out | 23.57 | **7.18** | 30.34 | 7.16 |
| In&Out+ InfinityGAN | **9.11** | 6.78 | **15.31** | **7.19** |

Table 3: **Inference speed up with parallel batching.** Benefit from the spatial independent generation nature, InfinityGAN achieves up to 7.20× inference speed up by with parallel batching at 8192×8192 pixels. The complete table can be found in Appendix P.

| Method | Parallel Batch Size | # GPUs | Inference Time (second / image) | Speed Up |
|---|---|---|---|---|
| StyleGAN2 | N/A | 1 | OOM | - |
| Ours | 1 | 1 | 137.44 | ×1.00 |
| | 128 | 8 | 19.09 | ×7.20 |

and texture are disentangled and modeled separately by $G_S$ and $G_T$. The figure also shows that $G_S$ can generate a diverse set of structures realized by different $\mathbf{z}_l$.

**User study.** We use two-alternative forced choice (2AFC) between InfinityGAN and other baselines on the Flickr-Landscape dataset. A total of 50 participants with basic knowledge in computer vision engage the study, and we conduct 30 queries for each participant. For each query, we show two separate grids of 16 random samples from each of the comparing methods and ask the participant to select "the one you think is more realistic and overall structurally plausible." As presented in Table 1, the user study shows an over 90% of preference favorable to InfinityGAN against all baselines.

### 4.2 ABLATION STUDY: THE POSITIONAL INFORMATION IN GENERATOR

As discussed in Section 3.3, we hypothesize that StyleGAN2 highly relies on the positional information from the zero-paddings. In Table 1 and Figure 4, we perform an ablation by removing all paddings from StyleGAN2+NCI, yielding StyleGAN2+NCI+PFG that has no positional information in the generator. The results show that StyleGAN2+NCI+PFG fails to generate reasonable image structures, and significantly degrades in all FID settings. Then, with the proposed $G_S$, the positional information is properly provided from $\mathbf{z}_S$, and resumes the generator performance back to a reasonable state.

### 4.3 APPLICATIONS

**Spatial style fusion.** Given a single global latent variable $\mathbf{z}_g$, the corresponding infinite-pixel image is tied to a single modal of global structures and styles. To achieve greater image diversity and allow the user to interactively generate images, we propose a spatial fusion mechanism that can spatially combine two global latent variables with a smooth transition between them. First, we manually define multiple style centers in the pixel space and then construct an initial fusion map by assigning pixels to the nearest style center. The fusion map consists of one-hot vectors for each pixel, forming a style assignment map. According to the style assignment map, we then propagate the styles in all intermediate layers. Please refer to Appendix L for implementation details. Finally, with the fusion maps annotated for every layer, we can apply the appropriate $\mathbf{z}_g$ from each style center to each pixel using feature modulation.Note that the whole procedure has a similar inference speed as the normal synthesis. Figure 7 shows synthesized fusion samples.

**Outpainting via GAN Inversion.** We leverage the pipeline proposed in In&Out (Cheng et al., 2021) to perform image outpainting with latent variable inversion. All loss functions follow the ones proposed in In&Out. We first obtain inverted latent variables that generates an image similar to the given real image via GAN inversion techniques, then outpaint the image by expanding $\mathbf{z}_l$ and $\mathbf{z}_n$ with their unit Gaussian prior. See Appendix K for implementation details.

In Table 2, our model performs favorably against all baselines in image outpainting (Boundless (Teterwak et al., 2019), NS-outpaint (Yang et al., 2019), and In&Out (Cheng et al., 2021)).

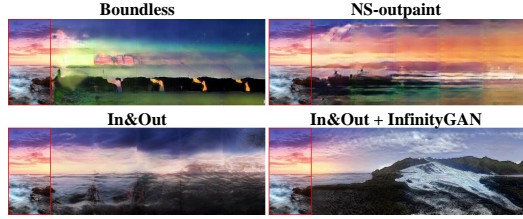

Figure 8: **Outpainting long-range area.** InfinityGAN synthesizes continuous and more plausible outpainting results for arbitrarily large outpainting areas. The real image annotated with red box is 256×128 pixels.

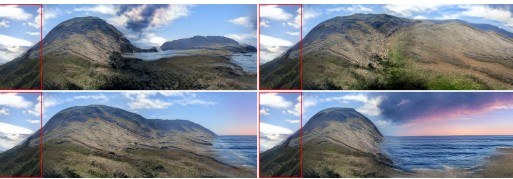

Figure 9: **Multi-modal outpainting.** InfinityGAN can natively achieve multi-modal outpainting by sampling different local latents in the outpainted region. The real image annotated with red box is 256×128 pixels. We present more outpainting samples in Appendix M.

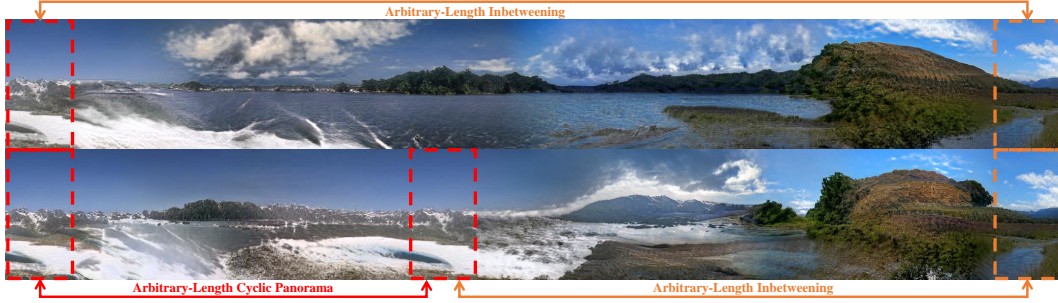

Figure 10: **Image inbetweening with inverted latents.** The InfinityGAN can synthesize arbitrary-length cyclic panorama and inbetweened images by inverting a real image at different position. The top-row image size is 256×2080 pixels. We present more samples in Appendix N and Appendix O.

As shown in Figure 8, while dealing with a large outpainting area (e.g., panorama), all previous outpainting methods adopt a sequential process that generates a fixed region at each step. This introduces obvious concatenation seams, and tends to produce repetitive contents and black regions after the multiple steps. In contrast, with InfinityGAN as the image prior in the pipeline of (Cheng et al., 2021), we can directly outpaint arbitrary-size target region from inputs of arbitrary shape. Moreover, in Figure 9, we show that our outpainting pipeline natively supports multi-modal outpainting by sampling different local latent codes in the outpainting area.

**Image inbetweening with inverted latent variables.** We show another adaptation of outpainting with model inversion by setting two sets of inverted latent variables at two different spatial locations, then perform spatial style fusion between the variables. Please refer to Appendix K for implementation details. As shown in Figure 10, we can naturally inbetween (Lu et al., 2021) the area between two images with arbitrary distance. A cyclic panorama of arbitrary width can also be naturally generated by setting the same image on two sides.

**Parallel batching.** The nature of spatial-independent generation enables parallel inference on a single image. As shown in Table 3, by stacking a batch of patches together, InfinityGAN can significantly speed up inference at testing up to 7.20 times. Note that this speed-up is critical for high-resolution image synthesis with a large number of FLOPs.

## 5    CONCLUSIONS

In this work, we propose and tackle the problem of synthesizing infinite-pixel images, and demonstrate several applications of InfinityGAN, including image outpainting and inbetweening.

Our future work will focus on improving InfinityGAN in several aspects. First, our Flickr-Landscape dataset consists of images taken at different FoVs and distances to the scenes. When InfinityGAN composes landscapes of different scales together, synthesized images may contain artifacts. Second, similar to the FoV problem, some images intentionally include tree leaves on top of the image as a part of the photography composition. These greenish textures cause InfinityGAN sometimes synthesizing trees or related elements in the sky region. Third, there is still a slight decrease in FID score in comparison to StyleGAN2. This may be related to the convergence problem in video synthesis (Tian et al., 2021), in which the generator achieves inferior performance if a preceding network (e.g., the motion module in video synthesis) is jointly trained with the image module.

## 6 ACKNOWLEDGEMENTS

This work is supported in part by the NSF CAREER Grant #1149783 and a gift from Snap Inc.

## 7 ETHICS STATEMENT

Our work follows the General Ethical Principles listed at ICLR Code of Ethics (https://iclr.cc/public/CodeOfEthics).

The research in generative modeling is frequently accompanied by concerns about the misuse of manipulating or hallucinating information for improper use. Despite none of the proposed techniques aiming at improving manipulation of fine-grained image detail or hallucinating human activities, we cannot rule out the potential of misusing the framework to recreate fake scenery images for any inappropriate application. However, as we do not drastically alter the plausibility of synthesis results in the high-frequency domain, our research is still covered by continuing research in ethical generative modeling and image forensics.

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

# Appendix Table of Contents

# A    COMPARISONS WITH CONCURRENT WORK

## A.1    ALIS (SKOROKHODOV ET AL., 2021)

ALIS is a concurrent work that achieves a similar application in infinite-pixel generation by iteratively inbetween pairs of anchor patches. Here, we discuss some of the critical differences.

**ALIS has limited ability in extending vertically.** InfinityGAN achieves both vertical and horizontal extension while maintaining a plausible holistic appearance. ALIS only presents horizontal extension in the paper. Vertically connecting anchors on any dataset used in their paper will produce invalid structures (e.g., layered landscapes periodically stacking in the sky). Therefore, vertical anchor connection is limited to certain datasets, such as pattern-like textures or satellite images.

**Differences in the problem formulation.** InfinityGAN directly models each infinite-pixel image with a shared global latent variable using an implicit function. In contrast, ALIS learns to inbetween two independent global latent variables. Furthermore, InfinityGAN can still achieve the inbetweening setup similar to ALIS with spatial style fusion. However, applying ALIS to synthesize images with a shared global context will lead to periodically repeating patches, as shown in Figure 13.

**InfinityGAN allows free-form anchor placements without training.** ALIS has to designate a constant relative position between the anchors before training starts. It is also non-trivial for ALIS to inbetween multiple anchors. Meanwhile, as shown in Figure 12, our training-free spatial style fusion allows placing *any* number of anchors at *any* place. However, the flexibility also comes with a trade-off, our spatial style fusion is not trained with adversarial learning, causing the synthesis performance not compatible to the regularly synthesized images. Following the ALIS evaluation protocol, we train our InfinityGAN on the ALIS LHQ dataset at $256{\times}256$ resolution, yielding an FID 11.82 with regular synthesis and an $\infty$-FID 19.22 with spatial style fusion. In contrast, ALIS shows negligible performance gap between FID (10.48) and $\infty$-FID (10.64). Such a result suggest a flexibility-accuracy trade-off between InfinityGAN and ALIS.

**ALIS generates blocky and high-frequency lattice artifacts.** ALIS adopts the generation-by-parts design from COCO-GAN. As shown in Figure 11, a critical consequence is creating the blocky and lattice artifacts between patches, since the inter-patch continuity is unstably maintained with adversarial learning. In contrast, InfinityGAN is an improved version of COCO-GAN that the inter-patch continuity is guaranteed with implicit-function and padding-free generator design.

**ALIS still suffers from content repetition.** The content repetition problem is discussed in the ALIS paper as a limitation. InfinityGAN addresses the issue with a local latent space and enforces the contribution of the local variables with a diversity loss (2) between $\mathbf{z}_l$ and $\mathbf{z}_S$. We demonstrate that InfinityGAN does not have the content repetition problem in Figure 31.

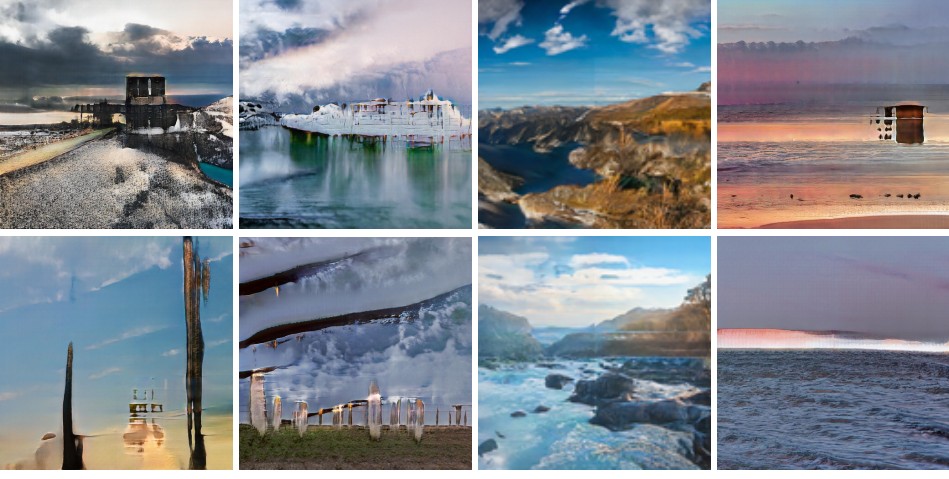

|  Blocky Artifacts  |  Discontinuity  |  Lattice Artifacts  |

Figure 11: **ALIS can suffer from blocky artifacts, inter-patch discontinuity and lattice artifacts.** We train ALIS with the official implementation at $1024{\times}1024$ resolution. We focus on the failure cases caused by COCO-GAN-based generation-by-parts framework, which artificially enforces the learned inter-patch continuity with adversarial learning.

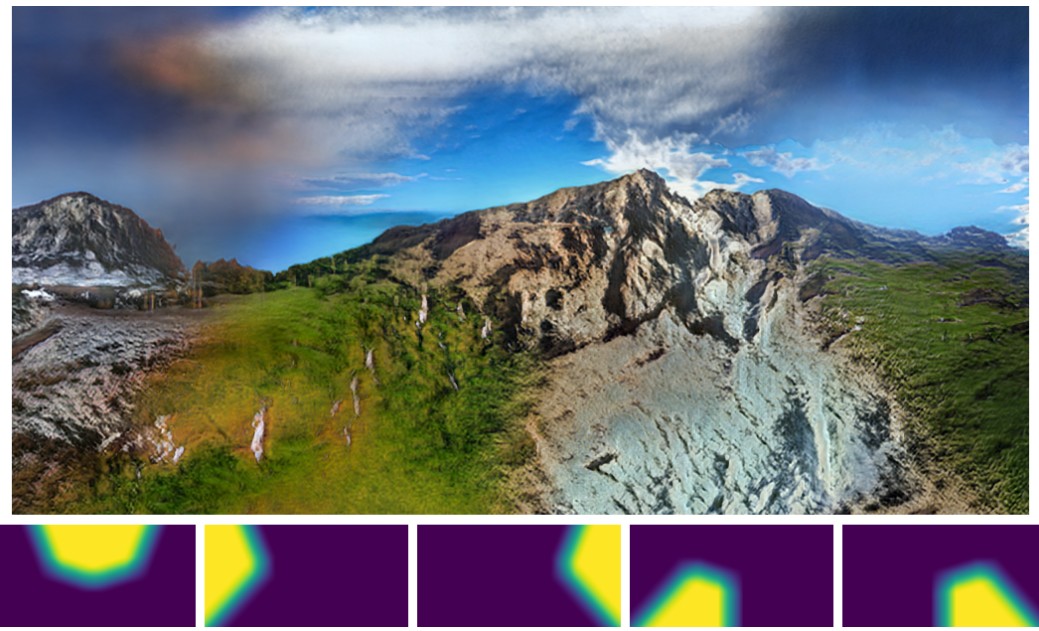

Figure 12: **InfinityGAN with free-form anchor placements.** The spatial style fusion of Infinity-GAN can place any number of style centers (called anchors in ALIS) at any location. We show case (top) a five-anchor example and (bottom) the corresponding regions covered by each style center.

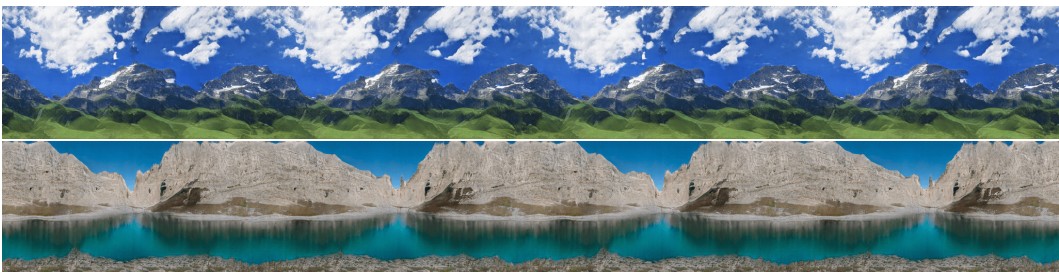

Figure 13: **ALIS cannot synthesis with a single holistic appearance.** ALIS synthesizes repetitive content if all anchors share the same latent vector.

## A.2 MS-PIE (XU ET AL., 2021).

MS-PIE found the positional information is crucial for GANs training. The paper explores different positional encoding schema, including sinusoidal coordinate encoding and padding removal. We further discuss the relation and distinction between MS-PIE and InfinityGAN in these two modules.

**Extensible coordinate encoding.** The "expand" configuration of MS-PIE allows coordinate value extrapolation in the spatial dimension. However, its training sticks to a fixed coordinate matrix for each synthesis scale, and the framework aligns the real images to the fixed coordinate matrix with bilinear interpolation [1] during training. Such a design causes the synthesized content to attach to the coordinate matrix, thus inevitably creates repetitive content when the coordinate value periodically repeats as the matrix expanding. In Figure 14, we train a no-padding StyleGAN2 model with MS-PIE-expand setting (i.e., config (k) in Table 5 of MS-PIE paper) on our Flickr-Landscape dataset. The model is trained at scales 256, 384 and 512, then test at scale 1408.

---

[1]The corresponding official implementation: `https://github.com/open-mmlab/mmgeneration/blob/95f962e54815b8f72c015c134cd597e9eff3de36/mmgen/models/gans/mspie_stylegan2.py#L115`.

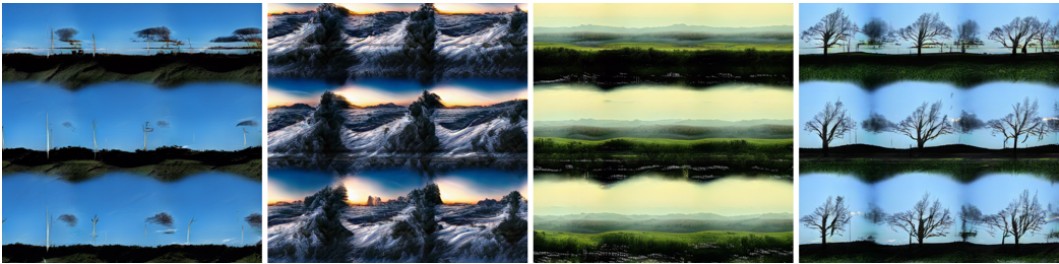

Figure 14: **MS-PIE creates repetitive content in the expand setting.** We train the no-padding StyleGAN2 with MS-PIE at 256, 384 and 512 scales, then synthesize at 1408 scale.

**Distinctions between no-padding and padding-free generator.** The no-padding generator (NPG) in MS-PIE is conceptually related to our padding-free generator (PFG), but fundamentally different while considering how the information of border pixels in the feature space is processed. NPG gradually loses the information from border pixels after each convolution, since the border pixels are less visited than the other pixels (consider a 3×3 convolutional kernel, most pixels will be scanned for 9 times, while edge pixels will be only scanned for 4 times, and corner pixels only 1 time). Further note that the information loss worsens exponentially as the network stacking more convolutional layers. In contrast, our PFG pads feature values from the neighbor context before the no-padding convolution is applied. The information loss caused by the no-padding convolution is a natural way to discard the information that is spatially too far away from and less related to the current context.

## B    CONCEPTUAL COMPARISONS AMONG DIFFERENT TASKS

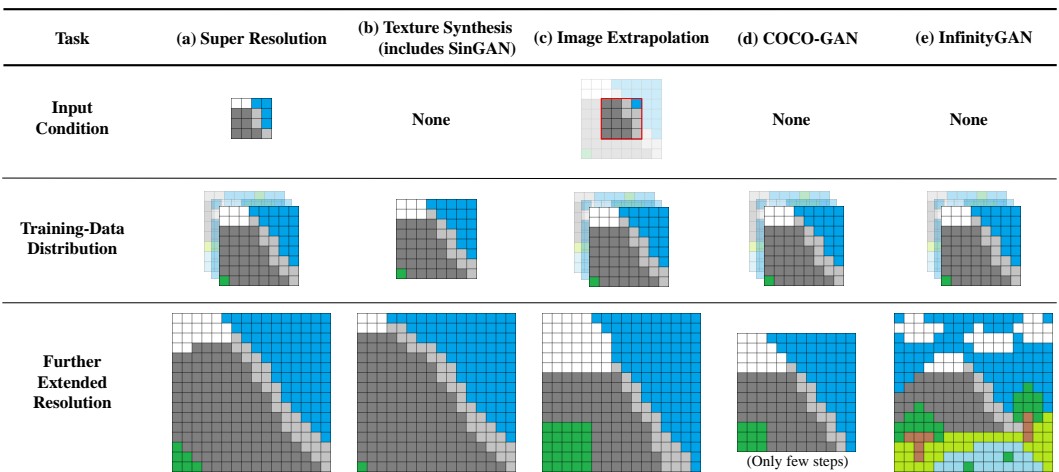

Figure 15: (a) **Super Resolution**: The final outputs inherit the coarse structure from and share the same field-of-view with the original input condition. (b) **Texture Synthesis**: Due to coordinate encoding, objects are generated near image the border, and the center of the image is filled with repetitive textures. (c) **Image Extrapolation**: Current extrapolation models tend to copy-and-paraphrase the conditional input or create mottled textural samples, leading to repetitive results especially when the outpainted region is large. (d) **COCO-GAN**: COCO-GAN can only synthesize samples slightly larger than its training distribution. (e) **InfinityGAN**: Ours InfinityGAN can synthesize a more favorable global structure at arbitrary resolutions without an input condition.

## C    IMPLEMENTATION DETAILS OF COORDINATES

We first derive the receptive field size $R$ of an $L$-layer $G_S$ after adding the $7{\times}7$ feature unfolding to all layers. Assume that the size of $\mathbf{z}_S$ (i.e., output of $G_S$, and input of $G_T$) is $M$. The value of $R$ can be derived as $M + (2{\times}3){\times}L$, where 3 is the half-size of the feature unfolding area. In practice, with

an architecture shown in Figure 18 with training patch size $101 \times 101$, we have $M = 11$, $L = 4$, and $R = 35$.

**Horizontal direction.** In order to avoid the generator discovering and exploiting any property of the periodic coordinate, we use a period $T$ much larger than $R$. Meanwhile, the period should be short enough to avoid the differences between consecutive coordinates vanish to almost zero. In practice, we use $T = 4 \times R$ for both of the cosine and sine coordinates.

**Vertical direction.** We utilize the property of the tanh function that its slope rapidly saturates to nearly zero and its value range is bounded by $[-1, 1]$. Such a property is aligned to the property of real-world scenery—the structural interactions among landscape objects are mostly concentrated around the horizon and rapidly saturates to a single modal (i.e., sky, ground, or water) in the vertical direction.

There are two hyperparameters while constructing the tanh coordinates: (i) a pair of cutoff points, and (ii) the sampling period of $\mathbf{z}_l$ in the spatial dimension.

- **Cutoff points** ($c_{\text{cut}}$) are a pair of values that, during training, we only sample the coordinates between the value pair. Such a hyperparameter is required since we are unable to sample all coordinates from an infinitely large coordinate system within the finite training steps. In practice, we set the cutoff points at $\pm 0.995$ of tanh-projected coordinate. Note that the underlying effect of using different cutoff point values is not well-investigated, we do not observe obvious changes with slightly different values.
- **Sampling period** ($d_l$) defines the distance between two spatially consecutive values of $\mathbf{z}_l$ are sampled within a training sample. Such a value relates to the grain of the representation that the generator overall models. It is equivalent to the occupation ratio of $\mathbf{z}_l$ between the cutoff points. Thus we can alternatively define a hyperparameter $V$ that defines the occupation ratio of $\mathbf{z}_l$ by $\frac{R}{R+V}$. Then, the sampling period can be derived with $d_l = 2 \times c_{\text{cut}} \times \frac{1}{R+V}$. In practice, we use $V = 10$, resulting in $d_l = 0.1\bar{3}$.

### C.1 MORE DISCUSSIONS ON THE CHOICE OF COORDINATE SYSTEM

In this paper, we mainly tackle the scenery image datasets, such as landscape, LSUN bridge and LSUN tower. The coordinate prior we introduced in Section 3.2 is specifically designed for such types of data, which has self-similarity in the horizontal direction and rapid mode-saturation in the vertical direction. However, we want to emphasize that the choice of coordinate system is a hyperparameter depending on the dataset. For instance, in Figure 16, we show that InfinityGAN can also work on satellite image dataset (Isola et al., 2017), which is more frequently used in texture synthesis task. A more natural choice of coordinate system in such a setting is using periodic coordinates in both horizontal and vertical directions. Nevertheless, in Figure 17 we observe that InfinityGAN can still produce visually plausible and globally sound appearances with using saturating (i.e., tanh) coordinates in the vertical direction and periodic coordinates in the horizontal direction.

In Table 4, we identify three categories of coordinate systems, but these options may not have covered all possibilities. One may need to be aware of the characteristics of the data and the intended outcome before selecting the appropriate coordinate system.

| Dataset Type | Object-centric | Texture | Scenery |
|---|---|---|---|
| Example | ImageNet, CelebA | Texture, satellite images | Landscape |
| Spatial Distribution of Content | N/A | Spatially agnostic | (Vertical) Spatially varying (Horizontal) Spatially agnostic |
| Horizontal Coordinate | Constant | Periodic (e.g., sin/cos) | Periodic (e.g., sin/cos) |
| Vertical Coordinate | Constant | Periodic (e.g., sin/cos) | Saturate (e.g., tanh) |

Table 4: **Possible coordinate designs for different types of dataset.** The choice of coordinate system for InfinityGAN is a dataset-dependent hyperparameter.

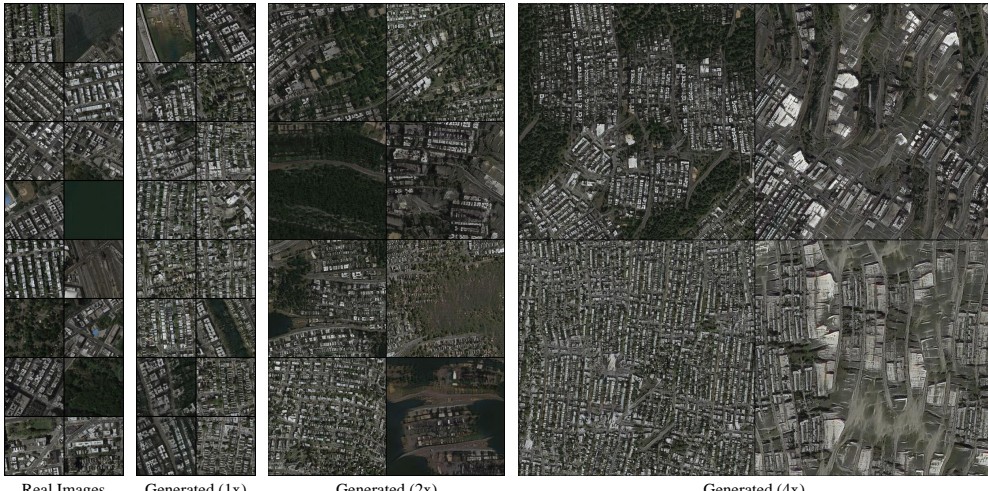

Figure 16: **Qualitative results on satellite image dataset.** InfinityGAN trained on satellite image dataset with cyclic coordinates on both vertical and horizontal directions.

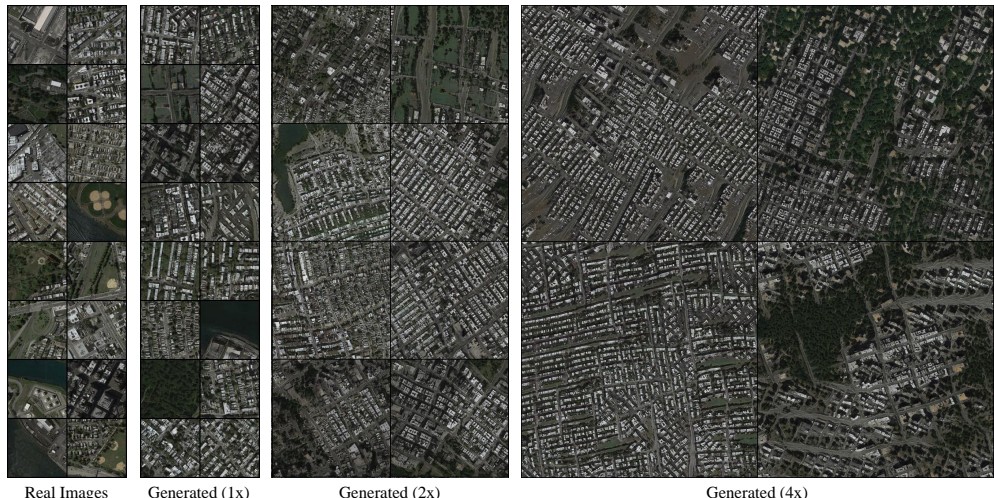

Figure 17: **Qualitative results on satellite image dataset.** InfinityGAN trained on satellite image dataset with saturating coordinates (i.e., tanh) on vertical direction and cyclic coordinates on horizontal direction.

# D    INFINITYGAN ARCHITECTURE

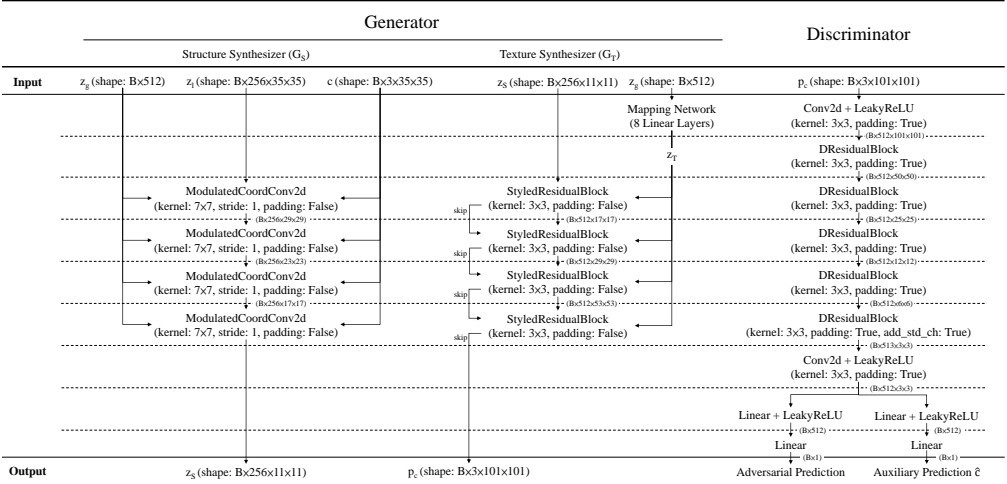

Figure 18: **A high-level overview of InfinityGAN model architecture.**

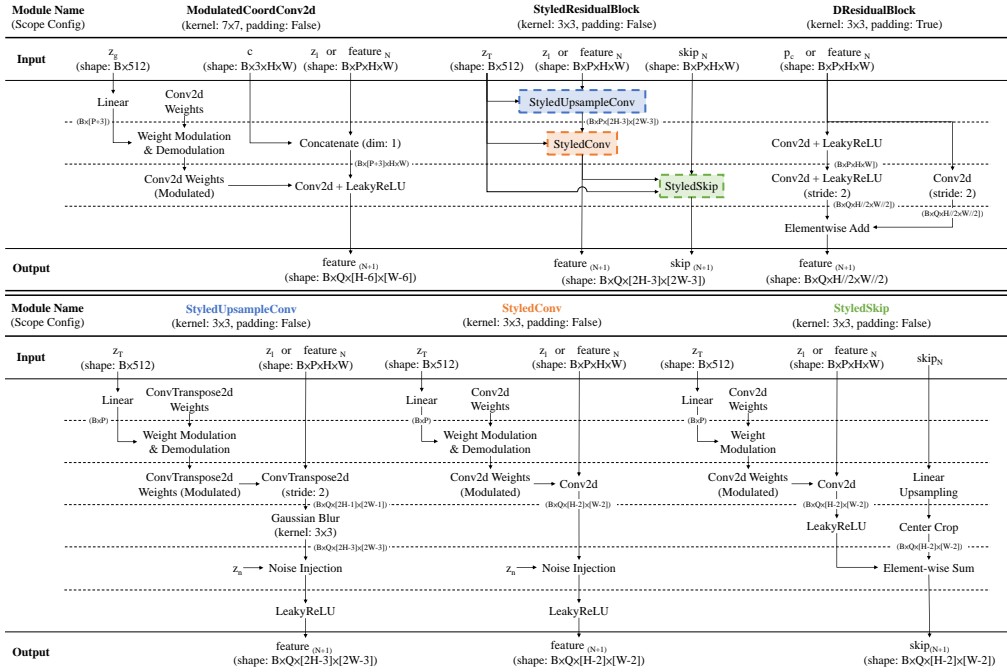

Figure 19: **The low-level design of each module within InfinityGAN.**

# E  MORE COMPARISON WITH BASELINES

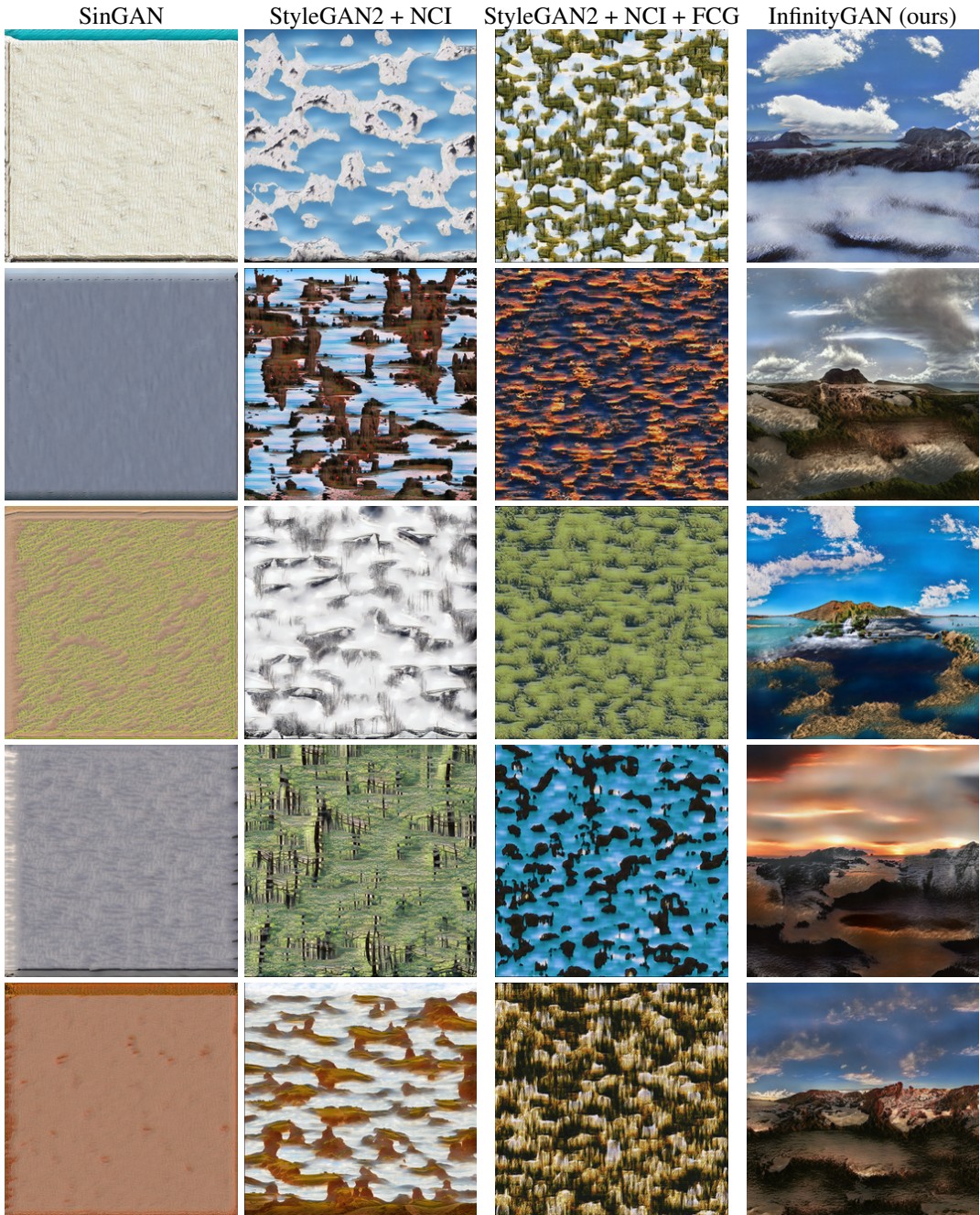

Figure 20: **More qualitative comparisons.** We show more samples on Flickr-Landscape at 1024×1024 pixels.

# F    MORE INFINITYGAN QUALITATIVE RESULTS

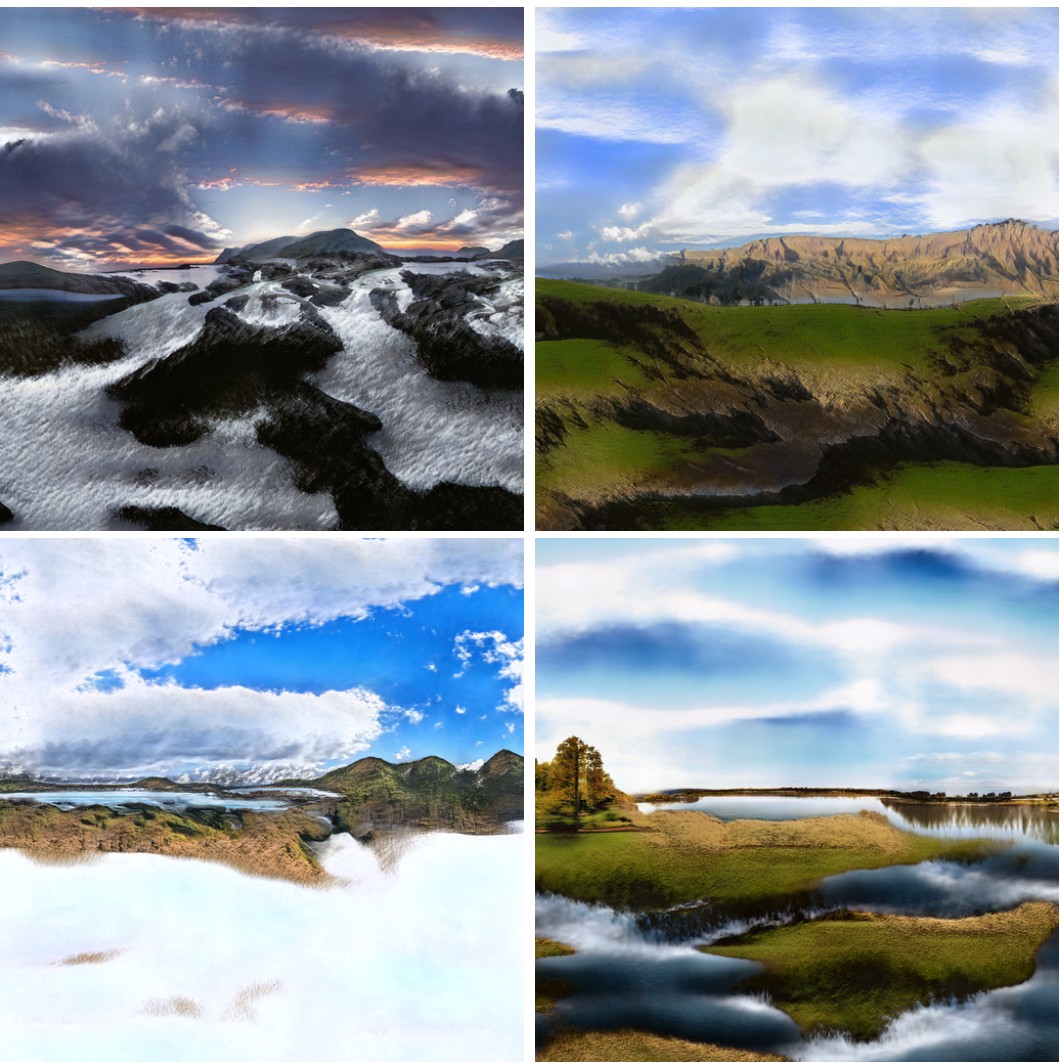

Figure 21: **More qualitative results.** We provide more images synthesized at 1024×1024 pixels with our InfinityGAN trained on Flickr-Landscape. All images are synthesized with the same model presented in the paper, which is trained with 101×101 patches cropped from 197×197 resolution real images. All images share the same coordinate and present a high structural diversity. Note that the images are down-sampled 2× to reduce file size.

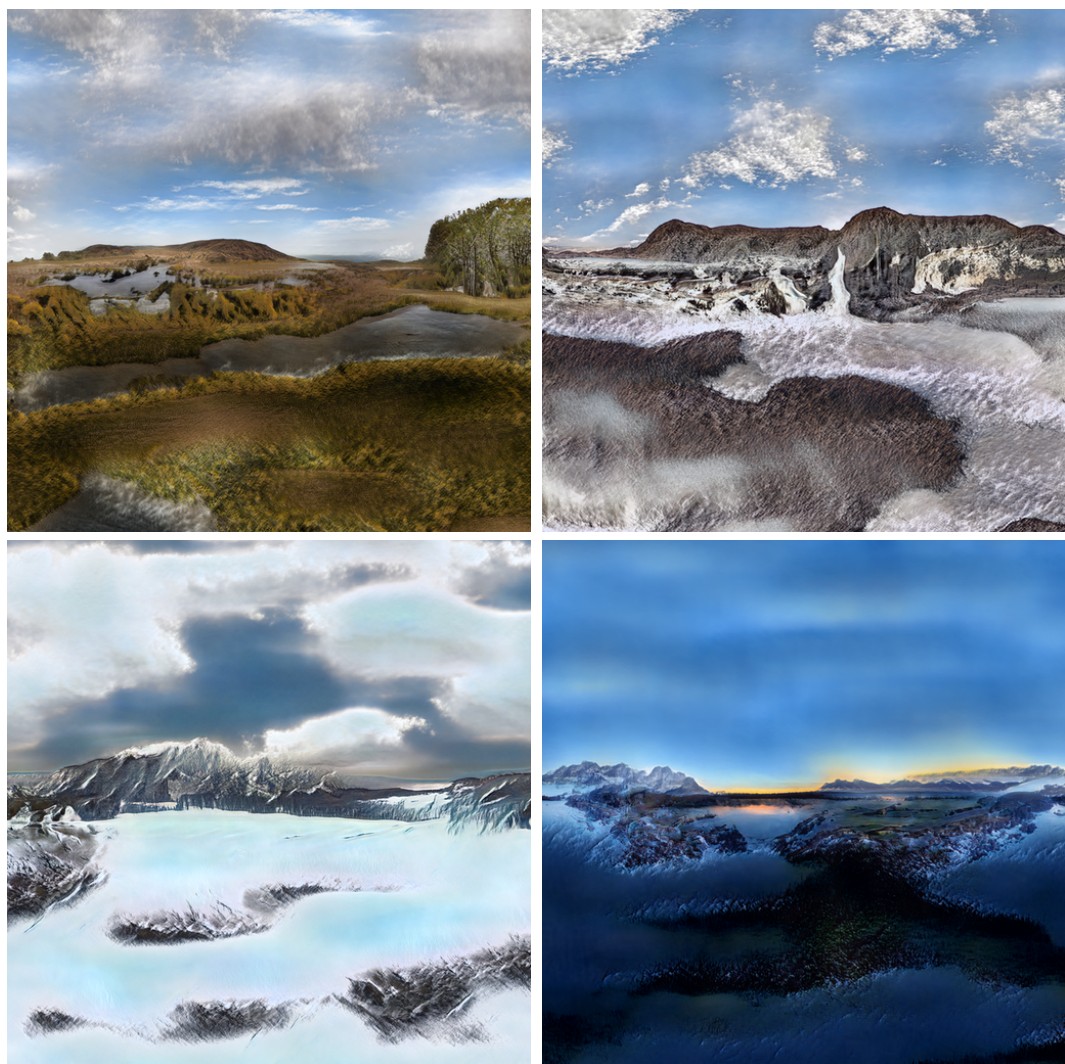

Figure 22: **More qualitative results.** We provide more images synthesized at 1024×1024 resolution with our InfinityGAN trained on Flickr-Landscape. All images are synthesized with the same model presented in the paper, which is trained with 101×101 resolution patches cropped from 197×197 resolution real images. All images share the same coordinate and present a high structural diversity. Note that the images are down-sampled 2× to reduce file size.

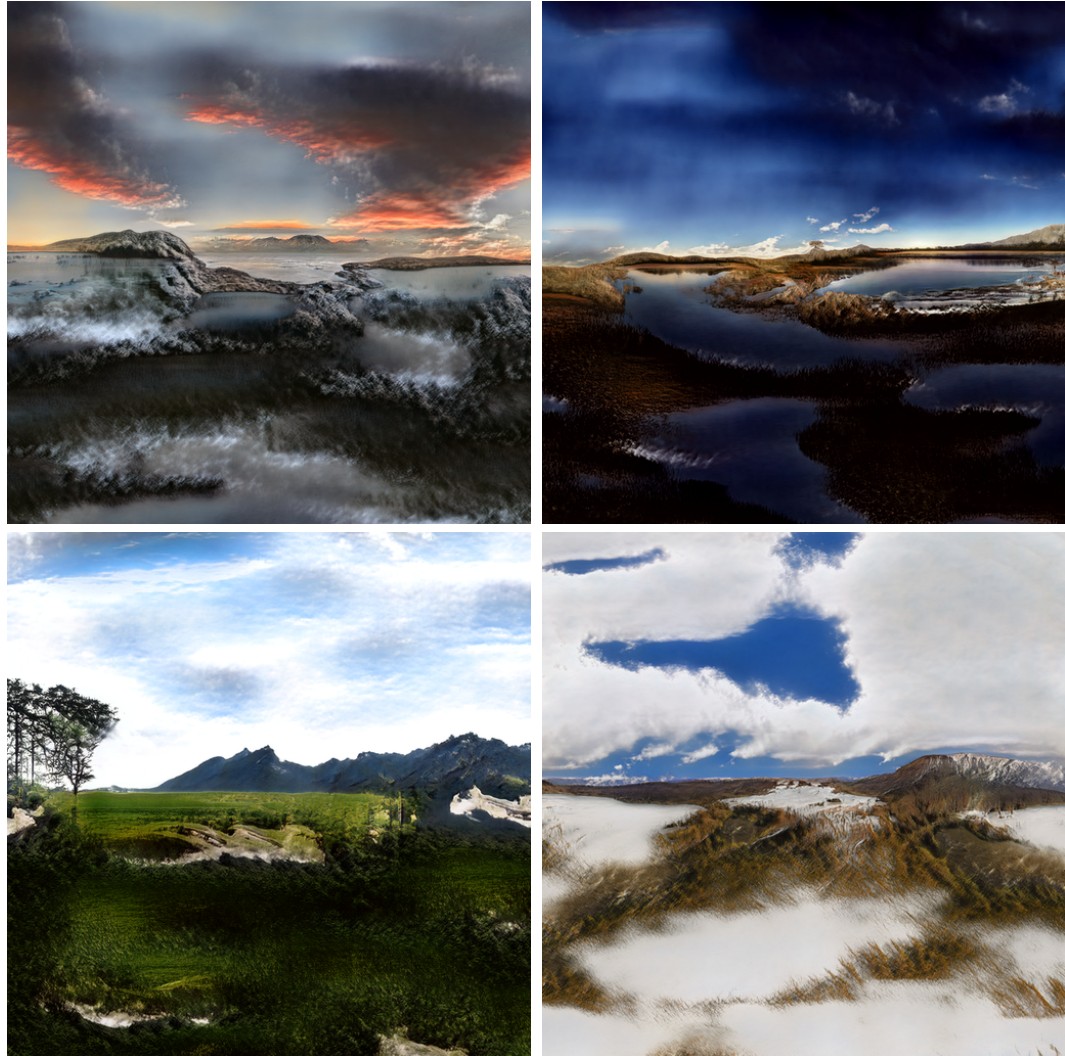

Figure 23: **More qualitative results.** We provide more images synthesized at 1024×1024 resolution with our InfinityGAN trained on Flickr-Landscape. All images are synthesized with the same model presented in the paper, which is trained with 101×101 resolution patches cropped from 197×197 resolution real images. All images share the same coordinate and present a high structural diversity. Note that the images are down-sampled 2× to reduce file size.

## G INFINITYGAN RESULTS AT A HIGHER RESOLUTION

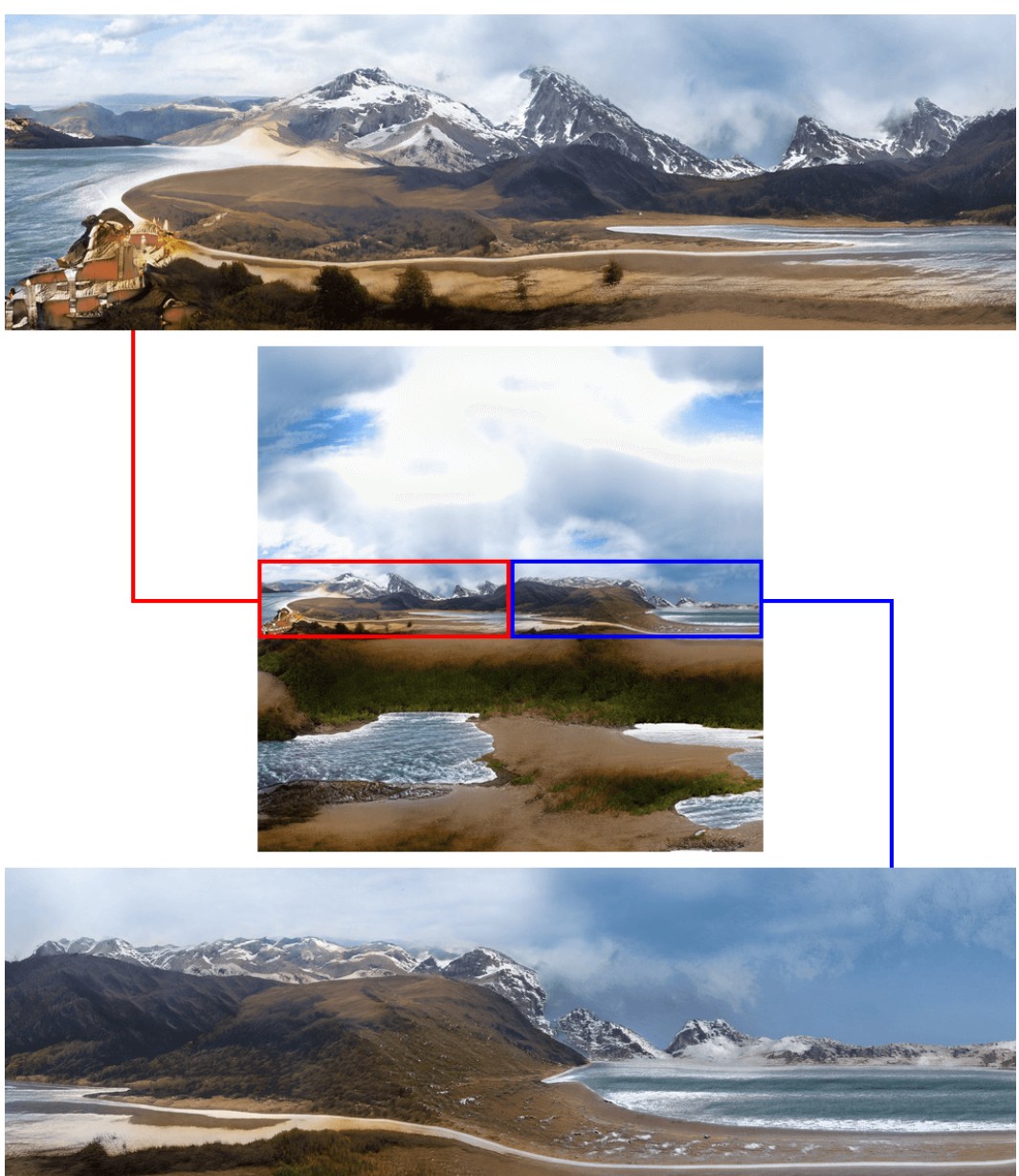

Figure 24: **InfinityGAN samples training at a higher resolution.** We synthesize 4096×4096 pixel images using InfinityGAN trained on Flickr-Landscape at 397×397 pixels patches cropped from 773×773 full images. The top and bottom rows are zoom-in view of the image. Note that the figure is 2× down-sampled to reduce file size.

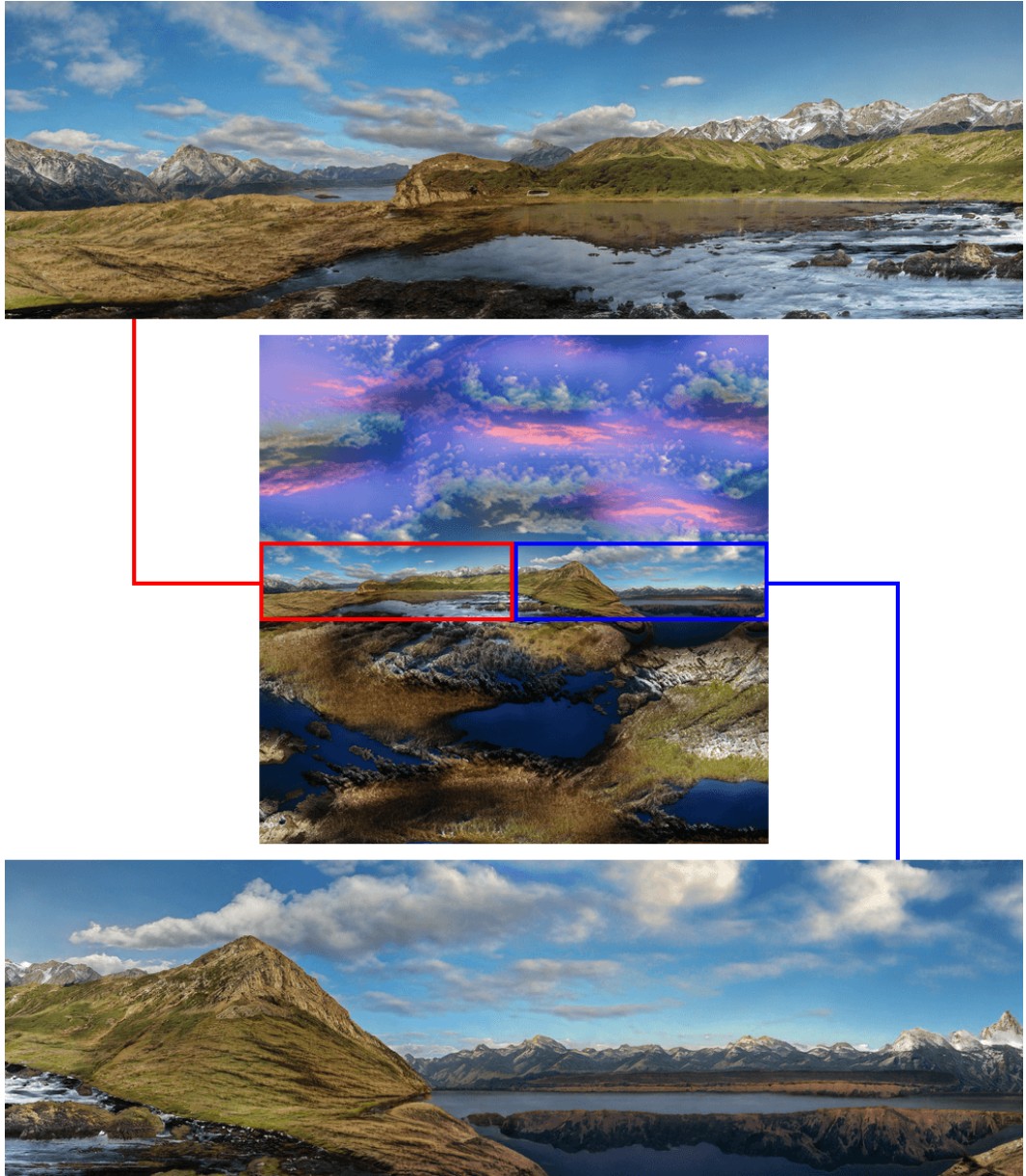

Figure 25: **InfinityGAN samples training at a higher resolution.** We synthesize 4096×4096 pixel images using InfinityGAN trained on Flickr-Landscape at 397×397 pixels patches cropped from 773×773 full images. The top and bottom rows are zoom-in view of the image. Note that the figure is 4× down-sampled to reduce file size.

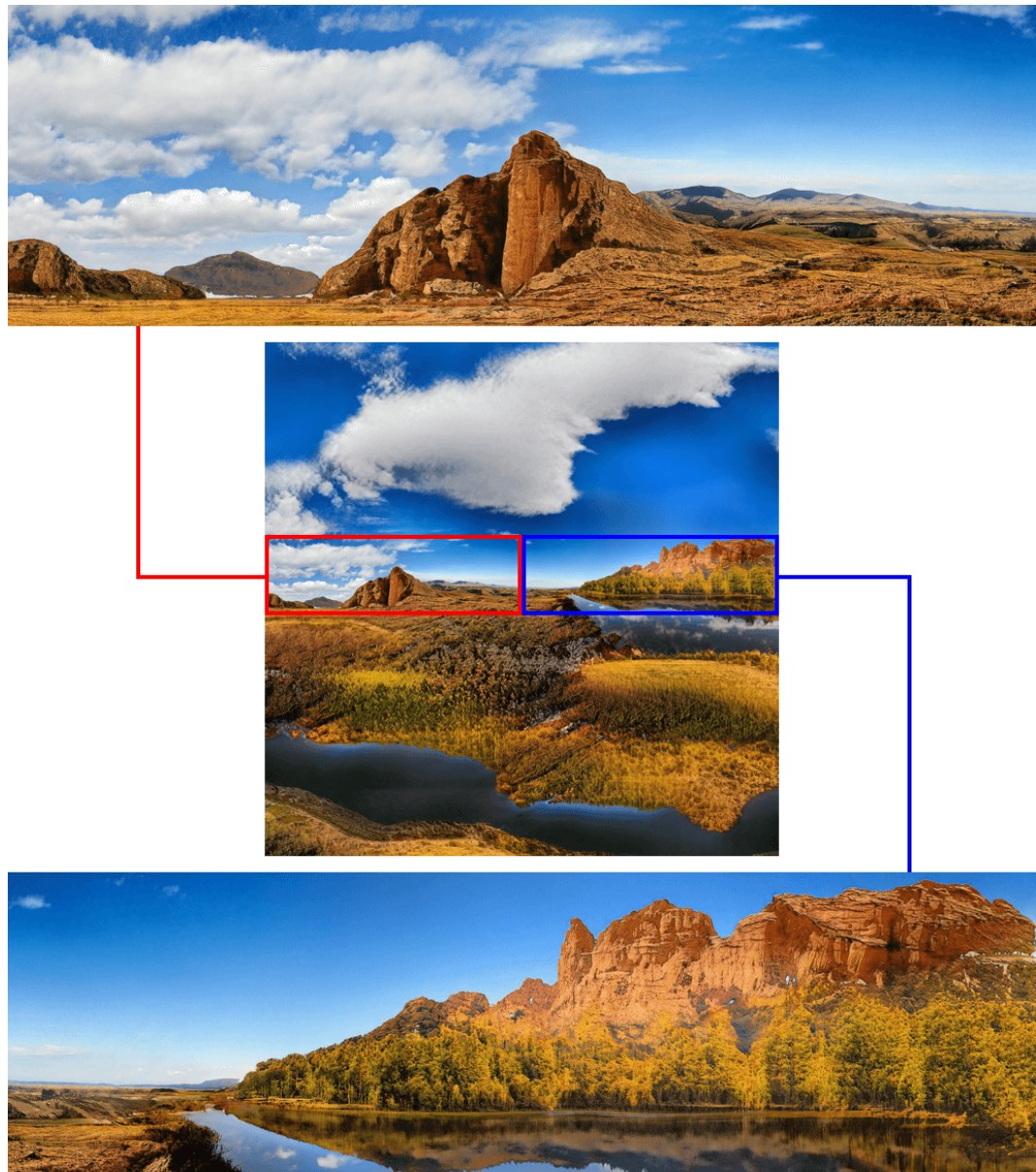

Figure 26: **InfinityGAN samples training at a higher resolution.** We synthesize 4096×4096 pixel images using InfinityGAN trained on Flickr-Landscape at 397×397 pixels patches cropped from 773×773 full images. The top and bottom rows are zoom-in view of the image. Note that the figure is 4× down-sampled to reduce file size.

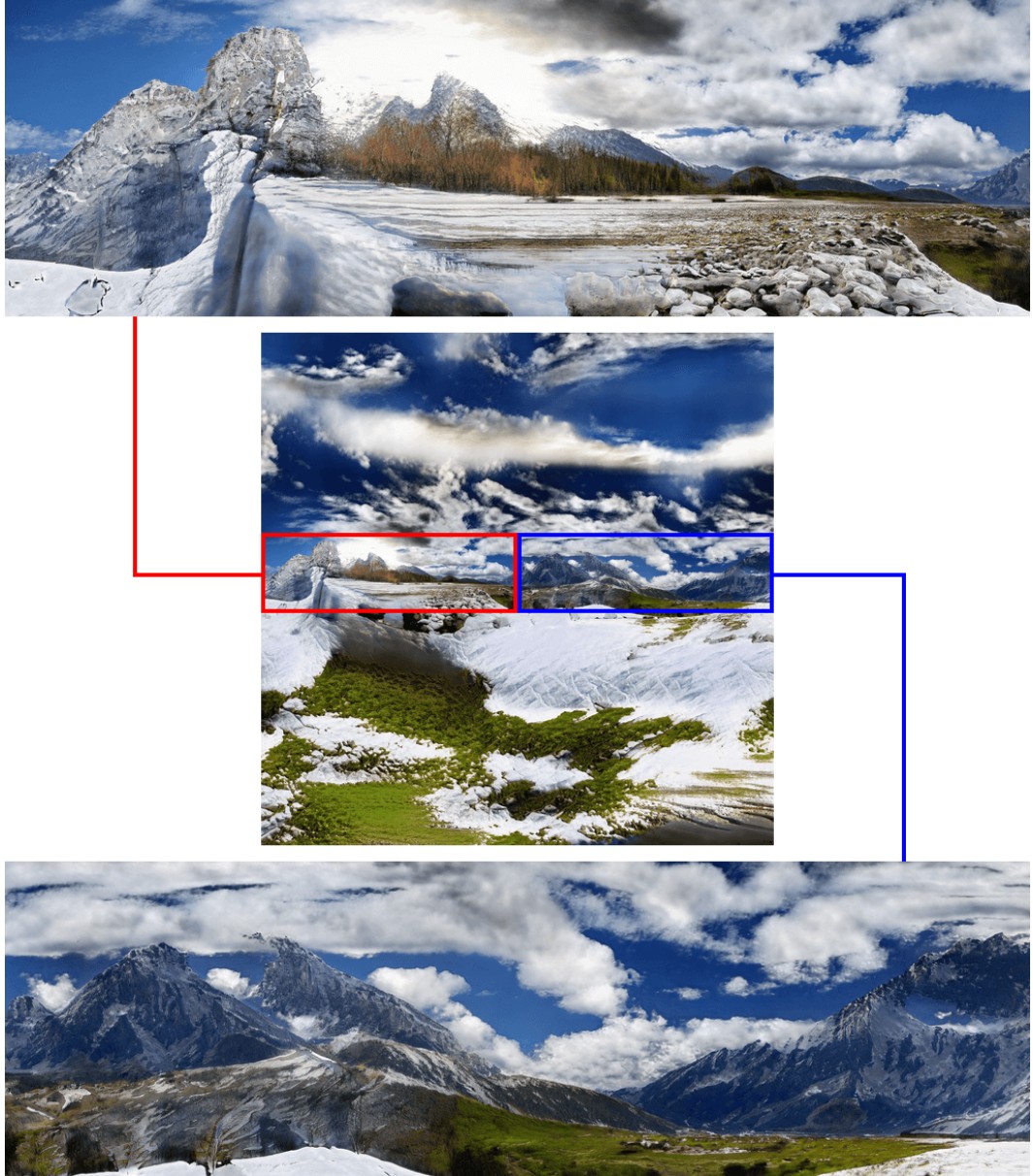

Figure 27: **InfinityGAN samples training at a higher resolution.** We synthesize 4096×4096 pixel images using InfinityGAN trained on Flickr-Landscape at 397×397 pixels patches cropped from 773×773 full images. The top and bottom rows are zoom-in view of the image. Note that the figure is 4× down-sampled to reduce file size.

# H   MORE RESULTS ON OTHER DATASETS

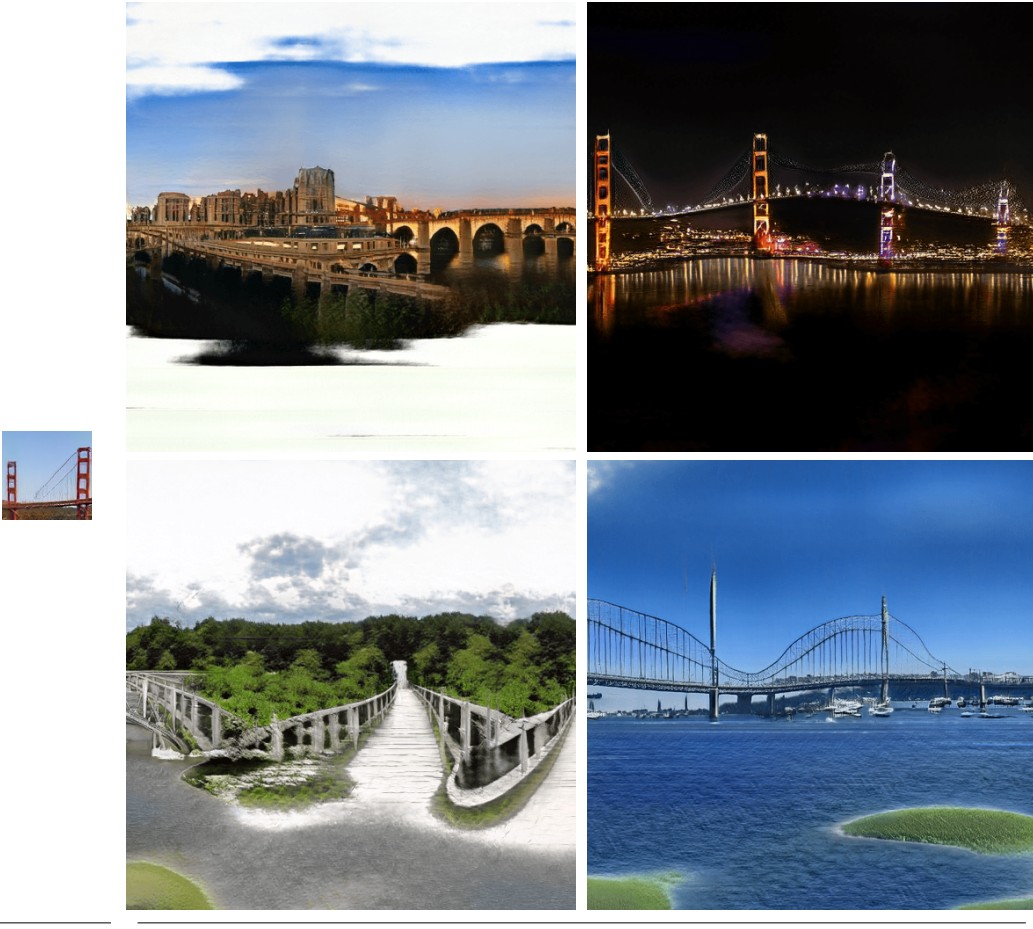

Training Size
(101×101)                 Test Size (512×512)

Figure 28: **LSUN bridge category.** InfinityGAN synthesis results at 512×512 pixels on LSUN bridge category. The model is trained with 101×101 pixels patches cropped from 197×197 resolution real images.

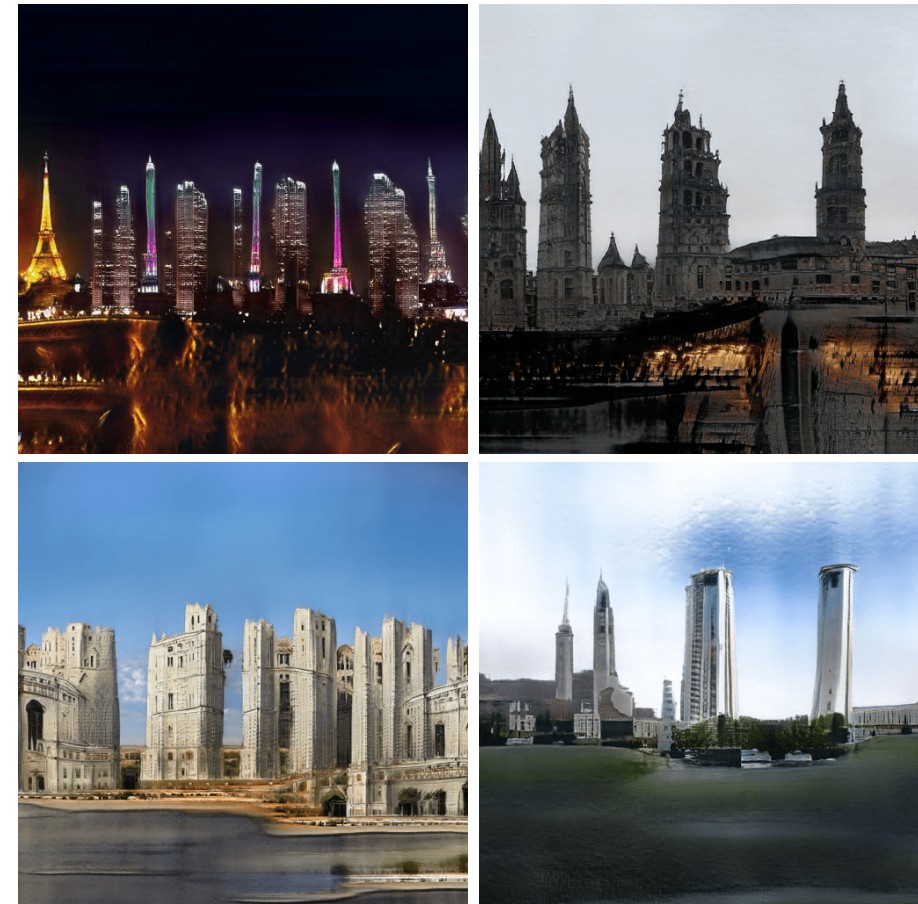

Training Size
(101×101)                   Test Size (512×512)

Figure 29: **LSUN tower category.** InfinityGAN synthesis results at 512×512 pixels on LSUN tower category. The model is trained with 101×101 pixels patches cropped from 197×197 resolution real images.

# I   MORE DIVERSITY VISUALIZATION

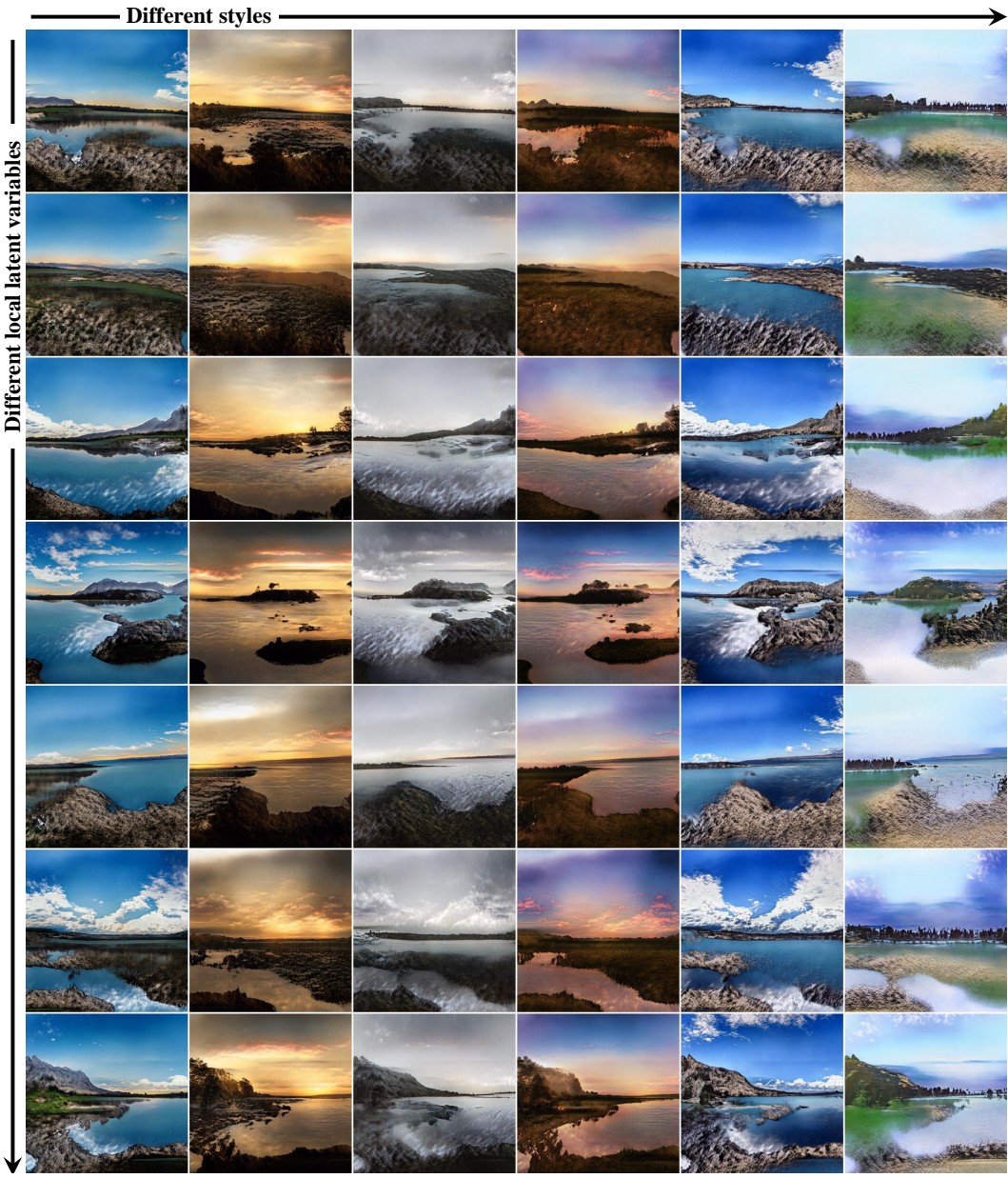

Figure 30: **Generation diversity.** We show that structure synthesizer and texture synthesizer separately models structure and texture by changing either the local latent or style while all other variables are fixed. The results also show that InfinityGAN can synthesize a diverse set of landscape structures at the same coordinate. All samples are synthesized at 389×389 pixels with InfinityGAN trained at 101×101.

## J    OUR BEST ATTEMPT IN INCLUDING AN EVERLASTING IMAGE

Figure 31: We provide a 256×9984 pixels sample synthesized with InfinityGAN. The sample shows that (a) our InfinityGAN can generalize to arbitrarily-large sizes, and (b) the synthesized contents do not self-repeat while using the sample global latent variable $\mathbf{z}_\mathrm{g}$.

## K    IMPLEMENTATION DETAILS OF IMAGE OUTPAINTING AND INBETWEENING VIA INVERSION

Our pipeline is similar to that of InOut (Cheng et al., 2021). With a given image $x$, the objective of GAN-model inversion is to recover a set of generator-parameter-dependent input latent variables $z^*$ that can synthesize a resulting image $x^*$ that is similar to $x$. There exist multiple different implementations to recover $z^*$, we adopt the gradient-descent-based method, which optimizes $z^*$ as a set of learnable parameters with carefully designed objective functions. In the context of InfinityGAN, we optimize four groups of variables: $\mathbf{z}_\mathrm{g}$, $\mathbf{z}_\mathrm{l}$, style ($\mathbf{z}_\mathrm{T}$) and $\mathbf{z}_\mathrm{n}$. In particular, we uses the $\mathcal{W}^+$ space

formulation (Abdal et al., 2020; Wulff & Torralba, 2020) for $\mathbf{z}_T$, named $\mathbf{z}_T^+$, where each layer in $G_T$ has its own set of $\mathbf{z}_T{}^i$ and each of the $\mathbf{z}_T{}^i$ is optimized separately.

**Objective functions.** We first introduce two image-distance losses to the inversion objectives:

$$
\begin{aligned}
\mathcal{L}_{\text{pix}} &= \|x - x^*\|_2\,, \\
\mathcal{L}_{\text{percept}} &= \text{LPIPS}(x^*, x)\,,
\end{aligned}
\tag{7}
$$

which LPIPS is the learned perceptual distance proposed by Zhang et al._ (Zhang et al., 2018).

We utilize the Gaussianized latent space technique proposed by Wulff and Torralba (Wulff & Torralba, 2020), which uses a LeakyReLU of slope 5 to discount the last activation function (a LeakyReLU with a slope 0.2) in the StyleGAN2 mapping layer, and recovers Gaussian-like marginal distribution of $\mathbf{z}_T^+$. With the Gaussian distribution prior, we can use the empirical mean $\mu_T^+$ and covariance matrix $\Sigma_T^+$ (computed by sampling 10,000 $\mathbf{z}_T^+$ via $\mathbf{z}_g$) to recover an estimated $\mathbf{z}_T^+$ distribution. With the empirical statistics, (Wulff & Torralba, 2020) proposes to compute Mahalanobis distance:

$$
d_M(z, \mu, \Sigma) = (z - \mu)^T\, \Sigma^{-1}\, (\mathbf{z}_T^+ - \mu).
\tag{8}
$$

Then, we construct a prior loss $\mathcal{L}_{\text{prior}}$ (Wulff & Torralba, 2020) that regularizes the $\mathbf{z}_g$, $\mathbf{z}_l$ and $\mathbf{z}_T^+$ with Mahalanobis distance:

$$
\mathcal{L}_{\text{prior}} = \lambda_\alpha d_M(\mathbf{z}_g, 0, \mathbf{I}) + \lambda_\beta d_M(\mathbf{z}_l, 0, \mathbf{I}) + \lambda_\gamma d_M(\mathbf{z}_T^+, \mu_T^+, \Sigma_T^+)\,,
\tag{9}
$$

which $\lambda_\alpha$, $\lambda_\beta$ and $\lambda_\gamma$ are weight factors. For the cases of $\mathbf{z}_g$ and $\mathbf{z}_l$ with zero means and unit variances, the prior loss degenerates to an $l_2$ loss.

Following StyleGAN2, we adopt a noise regularization loss $\mathcal{L}_{\text{nreg}}$ and noise renormalization. The full objective function of the inversion is:

$$
\mathcal{L}_{\text{inv}} = \lambda_{\text{pix}}\mathcal{L}_{\text{pix}} + \lambda_{\text{percept}}\mathcal{L}_{\text{percept}} + \mathcal{L}_{\text{prior}} + \lambda_{\text{nreg}}\mathcal{L}_{\text{nreg}}\,,
\tag{10}
$$

where $\lambda$s are the weighting factor of each loss terms.

**Hyperparameters.** For all tasks and datasets, we set $\lambda_{\text{pix}} = 10$, $\lambda_{\text{percept}} = 10$, $\lambda_{\text{nreg}} = 1{,}000$, $\lambda_\alpha = 10$, $\lambda_\beta = 10$, and $\lambda_\gamma = 0.01$. We use Adam (Kingma & Ba, 2015) optimizer with a learning annealing (Karras et al., 2020) from 0.1 to 0 for 1000 iterations. We use a batch size of 1 to avoid batch samples interfere with each other. Despite we observe batched inversion can sometimes yield superior results, it is not a conventional setting for real-world applications that batched inputs are mostly unavailable, it also significantly increases the stochasticity while reproducing the results.

**Outpainting and inbetweening with inverted latent variables.** With the inverted latent variables, we perform image outpainting by spatially extend the $\mathbf{z}_l^*$ with its unit Gaussian prior, while using $\mathbf{z}_g^*$ and $\mathbf{z}_T^{+*}$ everywhere. For image inbetweening, $\mathbf{z}_l^*$ is also extended with its unit Gaussian prior, while $\mathbf{z}_g^*$ and $\mathbf{z}_T^{+*}$ are fused with spatial style fusion. Notice that we do not optimize $c$ in our pipeline, since inverting $c$ is non-trivial and requires additional regularization losses. This introduces a limitation that the spatial position of the images is fixed after the inversion, the users have to perform the inversion optimization again if they want to assign $\mathbf{z}_l^*$ to a different location.

Another limitation is that, despite the use of the prior loss, some dimensions of the inverted latents tend to drift far away from the normal distribution. In combination with the use of the $\mathcal{W}^+$ space of $\mathbf{z}_T$, the inverted latents are of high-instability, sometimes introduce checkerboard-like artifacts, and frequently mix multiple irrelevant contexts together. We develop an interactive tool (released along with the code release) that allows users interactively and regionally resampling the undesired local latent variables $\mathbf{z}_l$ in the outpainting area. Such a limitation is highly related to the generalization of the inverted latents, we put it as an important future working direction.

## L    IMPLEMENTATION DETAILS OF SPATIAL STYLE FUSION

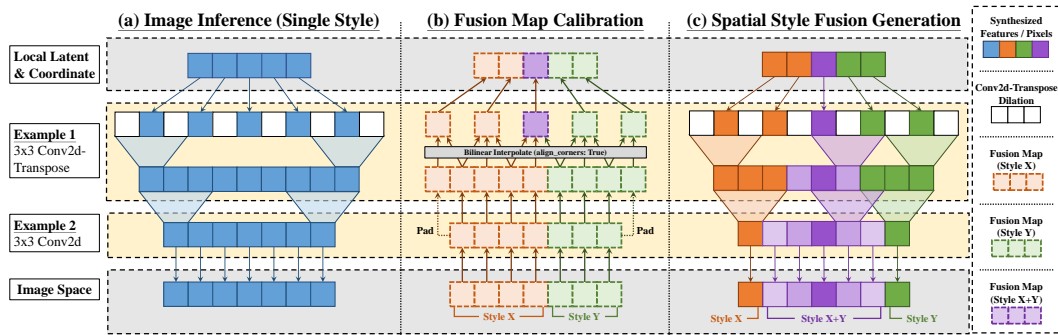

Figure 32: **Illustration of fusion map creation procedure and spatial fusion generation.** With a toy architecture example shown in (a) and a style fusion map in the pixel space (bottom of (b)), we can reversely create spatially aligned fusion maps in all intermediate layers by padding or interpolating the fusion map in the previous layer. Spatial style fusion in (c) uses the fusion maps to synthesize images with a natural style transition in the pixel space.

To achieve spatial style fusion in the existing InfinityGAN pipeline, we introduce two additional procedures: "style fusion map creation" and "fused modulation-demodulation". The former creates per-layer style fusion maps that specify the geometry of the style fusion area. The latter one is a modified version of feature modulation-demodulation that processes the volumetric styles created from the style fusion map.

**Style fusion map creation.** Given $N$ style centers designated by the user, in the pixel space, the target of style fusion map creation is to construct a set of style fusion maps for each layer of both $G_\mathrm{S}$ and $G_\mathrm{T}$. The fusion map is a spatially-shaped (i.e., batch$\times N \times H \times W$) tensor with $N$ channels that specifies the weight of the style for each spatial location, which the weights sums up to one across the $N$ dimension for each spatial position.

We first construct an initial fusion map in the pixel space by finding the spatially nearest style center, then assign a one-hot label for each spatial position in the initial fusion map. Since the spatial style fusion happens in all layers in the generator, we therefore reversely propagate the fusion map from the output of the generator to its input, we call such a procedure *fusion map calibration*. We show an illustration of fusion map calibration in Figure 32(b). The fusion map calibration starts from the image space and sequentially backward-constructs the fusion maps for all generator layers. For each pair (output-side and input-side) of the fusion maps in a network layer, we match the spatial dimension of the fusion map pair by padding or interpolating the output fusion map into a spatially aligned input fusion map. For different types of intermediate layers, the underlying implementation of the fusion map calibration can be slightly different, but a shared principle is to maintain a consistent geometrical position of the style fusion center throughout the generator.

In practice, such a binary map creates a sharp style transition that produces visible straight lines dividing the style regions. Accordingly, we apply a mean filter that smooths the style transition border. While different kernel sizes for the mean filter only alter the range of style transition and the visual smoothness, we use a kernel size of 127 in our experiments as it empirically produces good visual results.

**Fused modulation-demodulation.** After constructing the per-layer style fusion map, we can use the fusion maps to create volumetric styles (i.e., batch$\times D \times H \times W$) by weighted-sum the styles by the importance weights ($D$-channel dimension) in each spatial position. The volumetric styles are applied to each layer of the generator. As the feature modulation-demodulation strategy used in both StyleGAN2 and InfinityGAN is a pixel-wise operator, we can easily adapt it to volumetric styles. We demonstrate a possible implementation[2] of the fused modulation-demodulation in Figure 33.

---

[2]The forward function is based on the implementation from https://github.com/rosinality/stylegan2-pytorch.

Left (original StyleGAN2 forward function):

```python
import torch
import torch.nn.functional as F

def forward(self, feature, style):
    """
    feature: Feature with shape (B, C1, H, W)
    style  : Single style with shape (B, C2)
    """
    batch, in_c, in_h, in_w = feature.shape

    # Hyperparameters
    k = self.kernel_size    # Conv kernel size
    out_c = self.out_c      # Expected output channel
    rmpad = k // 2          # Zero-padding removal

    # Weight scaling (StyleGAN2)
    # Shape:
    # (1, ) * (out_c, in_c, k, k)
    # => (out_c, in_c, k, k)
    weight = self.scale * self.weight

    # Weight modulation (StyleGAN2)
    style = self.modulation(style)
    style = style.view(batch, 1, in_c, 1, 1)
    # Shape:
    # (1, out_c, in_c, k, k) * (batch, 1, in_c, 1, 1)
    # => (batch, out_c, in_c, k, k)
    weight = weight.unsqueeze(0) * style

    # Weight demodulation (StyleGAN2)
    demod = torch.rsqrt(
        weight.pow(2).sum([2,3,4]))
    weight *= demod.view(batch, out_c, 1, 1, 1)

    # Convolution
    feature = feature.view(1, batch*in_h, in_h, in_w)
    if self.upsample:
        weight = weight.view(batch, out_c, in_c, k, k)
        weight = weight.transpose(1, 2).reshape(
            batch*in_c, out_c, k, k)
        out = F.conv_transpose2d(
            feature, weight,
            padding=0, stride=2, groups=batch)

        # Clipping zero padding (ConvT special case)
        out = out[:, :, rmpad:-rmpad, rmpad:-rmpad]

        out = self.blur(out) # StyleGAN2 Gaussian blur
    else:
        weight = weight.view(batch*out_c, in_c, k, k)
        out = F.conv2d(
            feature, weight, padding=0, groups=batch)

    # Recover batch-channel shape due to grouping
    _, _, out_h, out_w = out.shape
    out = out.view(batch, out_c, out_h, out_w)

    return out
```

Right (spatial style fusion implementation):

```python
import torch
import torch.nn.functional as F

def fused_forward(self, feature, style):
    """
    feature: Feature with shape (B, C1, H, W)
    style  : Fusion style with shape (B, C2, H, W)
    """
    batch, in_c, in_h, in_w = feature.shape
    st_c = style.shape[1]

    # Hyperparameters
    k = self.kernel_size    # Conv kernel size
    out_c = self.out_c      # Expected output channel
    rmpad = k // 2          # Zero-padding removal

    # Weight scaling (StyleGAN2)
    # Shape:
    # (1, ) * (out_c, in_c, k, k)
    # => (out_c, in_c, k, k)
    weight = self.scale * self.weight

    # Weight modulation (Casted)
    #   The following two forms are equivalent:
    #    - conv(in=feature,      w=weight*style*demod)
    #    - conv(in=feature*style, w=weight) * demod
    #   StyleGAN2 uses the former one for speed.
    style = \
        style.permute(0, 2, 3, 1).reshape(-1, st_c)
    style = self.modulation(style)
    style = style.view(batch, in_h, in_w, in_c)
    style = style.permute(0, 3, 1, 2)
    feature = (style * feature) # (B, C, H, W)

    # Weight demodulation (Approximated)
    #   Feature demodulation use patch statistics.
    #   The approximation here is similar to a
    #   mean of statistics from all styles.
    demod = torch.zeros(batch, out_c, in_h, in_w)
    for i in range(in_h):
        for j in range(in_w):
            style_v = style[:, :, i, j] \
                .view(batch, 1, in_c, 1, 1)
            style_v = weight.unsqueeze(0) * style_v
            # style_v shape: (B, out_ch, in_ch, k, k)
            demod[:, :, i, j] = \
                torch.rsqrt(
                    style_v.pow(2).sum([2,3,4]))

    # Convolution
    # (All feature uses same weight, no need to group)
    if self.upsample:
        weight = weight.view(out_c, in_c, k, k)
        weight = weight.transpose(0, 1).contiguous()
        out = F.conv_transpose2d(
            feature, weight,
            padding=0, stride=2, groups=1)

        # Clipping zero padding (ConvT special case)
        out = out[:, :, rmpad:-rmpad, rmpad:-rmpad]

        # Late demodulation (match output shape)
        demod = F.interpolate(
            demod,
            size=(out.shape[-2], out.shape[-1]),
            mode="bilinear", align_corners=True)
        out = out * demod

        out = self.blur(out) # StyleGAN2 Gaussian blur
    else:
        out = F.conv2d(
            feature, weight, padding=0, groups=1)
        demod = demod[:, :, rmpad:-rmpad, rmpad:-rmpad]
        out = out * demod

    out = out.contiguous()

    return out
```

Figure 33: **Implementation of spatial style fusion.** We present (left) the original StyleGAN2 forward function, and (right) a corresponding implementation for the spatial style fusion. We align the related code blocks on the left and right.

## M  MORE QUALITATIVE RESULTS OF OUTPAINTING VIA INVERSION

Figure 34: **More outpainting via model inversion.** We present more outpainting results from InfinityGAN on Flickr-Scenery. We invert the latent variables from $256 \times 128$ pixels real images (marked with red box), then outpaint $256 \times 640$ area ($5 \times$ real image size).

## N  MORE QUALITATIVE RESULTS OF INBETWEENING VIA INVERSION

Figure 35: **More image inbetweening with model inversion.** By inverting the latent variables that reconstruct the two real images on two sides (marked with red box), InfinityGAN can naturally inbetween the two images arbitrarily distant away. We synthesize the 256×1280 images using InfinityGAN trained on Flickr-Scenery at 101×101 pixels.

## O    MORE QUALITATIVE RESULTS OF CYCLIC PANORAMIC INBETWEENING VIA INVERSION

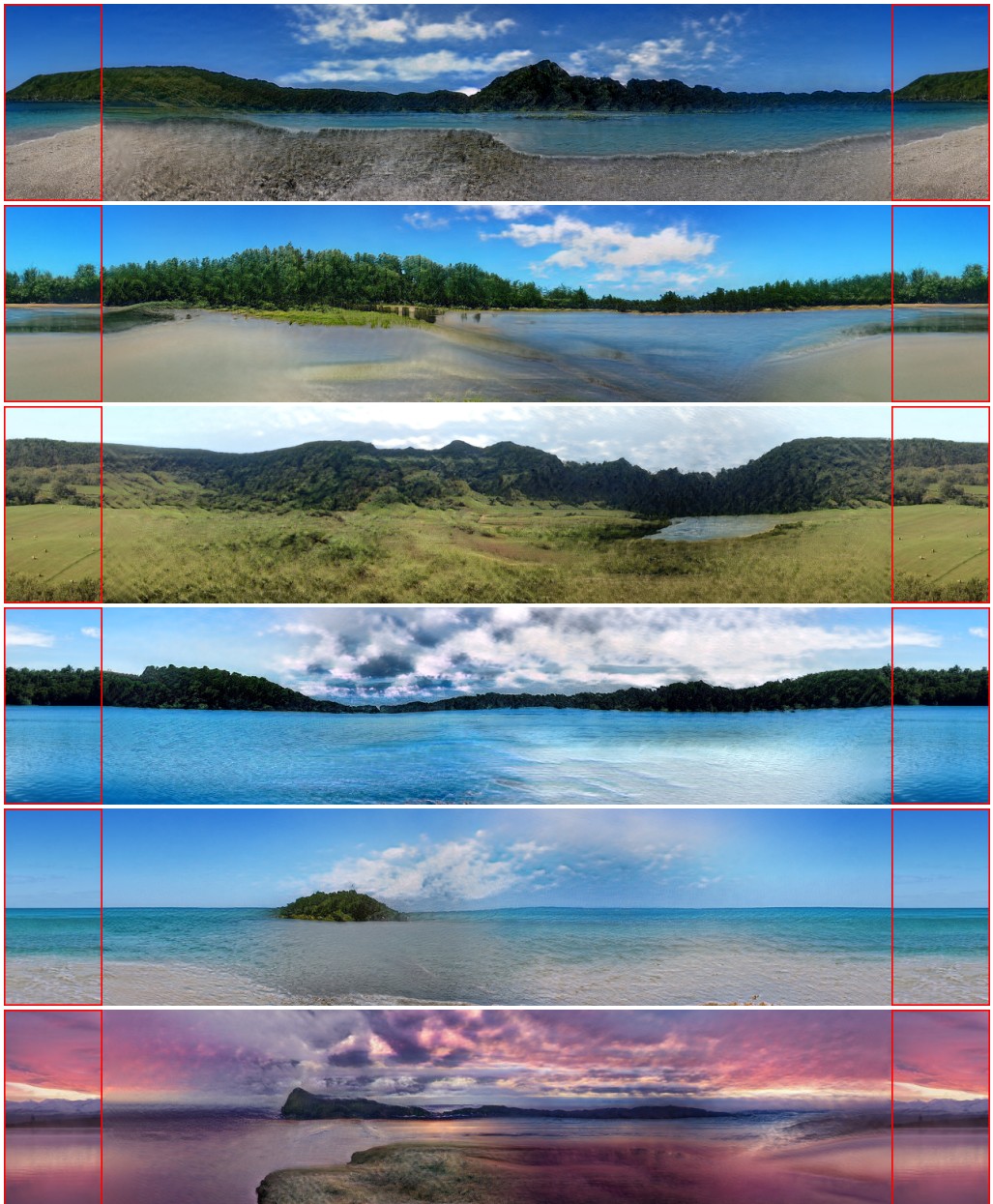

Figure 36: **More cyclic panorama synthesized with model inversion.** By setting the same real image on two sides (marked with red box) and inverting the latent variables that reconstruct the real image, InfinityGAN can naturally synthesize horizontally cyclic panoramic images with image inbetweening. We synthesize the 256×1280 images using InfinityGAN trained on Flickr-Scenery at 101×101 pixels.

## P EXPERIMENTAL DETAILS OF THE SPEED BENCHMARK WITH PARALLEL BATCHING

We perform all the experiments on a workstation with Intel Xeon CPU (E5-2650 2.20GHz) and 8 GTX 2080Ti GPUs. We implement our framework with Pytorch 1.6, and execute in an environment with Nvidia driver version 440.44, cuDNN version 4.6.5, and Cuda version 10.2.89.

We report the summation of pure GPU execution time and data scatter-collection time introduced by data parallelism. The model synthesizes a single image for each trial. We first warm up the GPUs with 10 proceeding trials without recording their statistics, then compute the mean and variance over 100 trials. The numbers are reported in Table 5.

Table 5: **Inference speed up with parallel batching.** Benefit from the spatial independent generation nature, InfinityGAN achieves up to 7.20× inference speed up by with parallel batching. We conduct all experiments at a batch size of 1, and OOM indicates out-of-memory. Note that the GPU time here accounts for pure GPU execution time and (if applicable) data-parallel scatter-aggregation time.

| Method | Generation Paradigm | Parallel Batch Size | # GPUs | GPU Time @ Inference Size (sec/image) | | | | Speed Up | MFLOPs |
| --- | --- | --- | --- | --- | --- | --- | --- | --- | --- |
| | | | | 1024×1024 | 2048×2048 | 4096×4096 | 8192×8192 | 8192×8192 | 1024×1024 |
| StyleGAN2 | One-Shot | - | 1 | 0.60 ± 0.01 | OOM | OOM | OOM | - | 6,642 |
| InfinityGAN (Ours) | One-Shot | - | 1 | 0.67 ± 0.01 | OOM | OOM | OOM | - | 6,815 |
| | Spatially Independent Generation | 1 | 1 | 1.24 ± 0.15 | 7.96 ± 0.17 | 34.35 ± 1.69 | 137.44 ± 1.85 | ×1.00 | 17,901 |
| | | 2 | | 1.58 ± 0.09 | 5.31 ± 0.13 | 24.13 ± 0.42 | 95.77 ± 1.63 | ×1.44 | |
| | | 4 | | 1.35 ± 0.01 | 5.20 ± 0.02 | 20.93 ± 0.04 | 82.52 ± 0.08 | ×1.67 | |
| | | 8 | | 1.28 ± 0.01 | 5.14 ± 0.02 | 19.63 ± 0.02 | 78.41 ± 0.17 | ×1.75 | |
| | | 16 | | 1.23 ± 0.01 | 5.01 ± 0.01 | 19.11 ± 0.02 | 76.41 ± 0.02 | ×1.80 | |
| | | 32 | 2 | 0.96 ± 0.01 | 3.90 ± 0.02 | 14.84 ± 0.06 | 59.33 ± 0.15 | ×2.32 | |
| | | 64 | 4 | 0.56 ± 0.01 | 2.25 ± 0.05 | 8.64 ± 0.11 | 35.20 ± 0.39 | ×3.90 | |
| | | 128 | 8 | 0.32 ± 0.05 | 1.30 ± 0.05 | 4.82 ± 0.06 | 19.09 ± 0.16 | ×7.20 | |

## Q ABLATION: FEATURE UNFOLDING

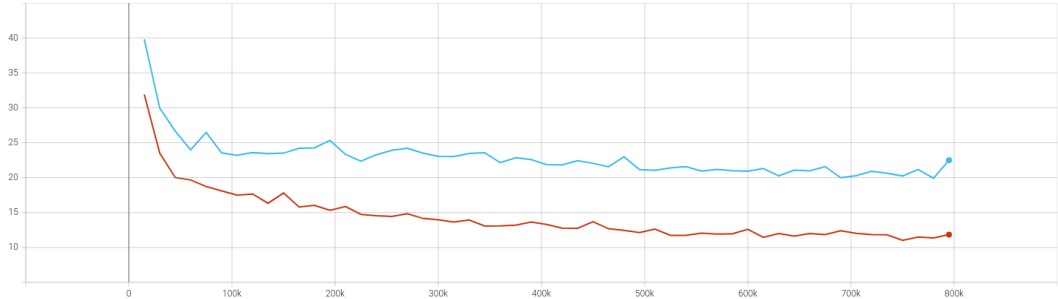

Figure 37: We plot the FID curve for a training episode of our complete InfinityGAN (red curve) and InfinityGAN without feature unfolding (blue curve). We observe the FID saturates at early stage if without feature unfolding.

## R    LIMITATIONS

Here we discuss some empirical limitations of InfinityGAN.

**Patch-training leads to performance degradation.** In order to construct a better conditional distribution in the vertical direction with an auxiliary loss $\mathcal{L}_{ar}$, InfinityGAN trains the generator with patches instead of full images. However, training with patches reduces the training field-of-view, thus leading to an inferior performance at the same field-of-view compared to a model trained with full images. Designing unsupervised mechanisms in learning $\mathcal{L}_{ar}$ without patch-cropping can improve InfinityGAN performance.

**Long-range coherence.** InfinityGAN assumes local coherence following a shared holistic appearance can achieve visually plausible synthesis. However, certain physical relationships still require long-range dependency, such as we can observe two suns in the top-left image in Figure 23, and twilight should only happen near the horizon in Figure 25. Despite InfinityGAN can independently sample pixels arbitrarily distant away, it remains unclear how to construct unsupervised losses for such conditions, as we only have finite-pixel images in training data.

**Strip-shaped artifacts.** We observe InfinityGAN creates a unique type of artifact that forms a strip-shaped structure sweeping through the sky or ground for a long distance, such as the clouds in the bottom-two image in Figure 23. We hypothesize the root cause of such an artifact is that the model attends too much to the strong structural characteristics of the horizon line and accidentally shares the representation with other less related contexts. We anticipate improving the modulation or coordinate encoding mechanisms may help suppress such behavior.

## S    IMPLEMENTATION ILLUSTRATION OF SCALEINV FID

```python
import torch.nn.functional as F

def eval_scaleinv_fid(real_imgs, fake_imgs, scale):
    # real_imgs: tensor of real images, shape (B, C, H, W).
    # fake_imgs: tensor of fake images, shape (B, C, H, W).
    # scale:     the scale of the current ScaleInv FID.
    fake_images = F.interpolate(
        fake_images,
        scale_factor=1/scale,
        mode="bilinear",
        align_corners=True)

    # The regular FID evaluation
    return eval_fid(real_images, fake_images)
```

## T    MORE COMPARISONS WITH TEXTURE SYNTHESIS METHOD

As we discussed in Section 1 that texture synthesis models are not directly applicable to real-world image synthesis. In Figure 38, we demonstrate such a problem by running TileGAN (Frühstück et al., 2019) on our Flickr-Landscape dataset. The results show that the random texture synthesis cannot produce plausible global structures. It is important to note that the images are shown in Tile-GAN and other texture synthesis papers (Bergmann et al., 2017; Jetchev et al., 2018) with plausible global structure, such as the satellite map of Jurassic Park and the paintings in TileGAN, are all conditioned on an image that gives the blueprint of the global structure.

**Implementation details of TileGAN experiment.** We use the officially released TileGAN pipeline to synthesize the results with randomly sampled latent variables. We follow the instructions and train a PGGAN model at 256×256 resolution, then test the model to synthesize at 512×512, 1024×1024, and 2048×2048 pixels. Note that we discovered that TileGAN alters the PGGAN architecture from residual-based ToRGB branch [3] to single ToRGB projection [4]. Such a modification is not described in the TileGAN paper, but can be found by diagnosing the model checkpoints released by the authors. The modification leads to significant visual-quality degradation. However, even without the visual-quality degradation, the lack of structural clues makes TileGAN impossible to infer a coherent global structure while synthesizing at larger image sizes.

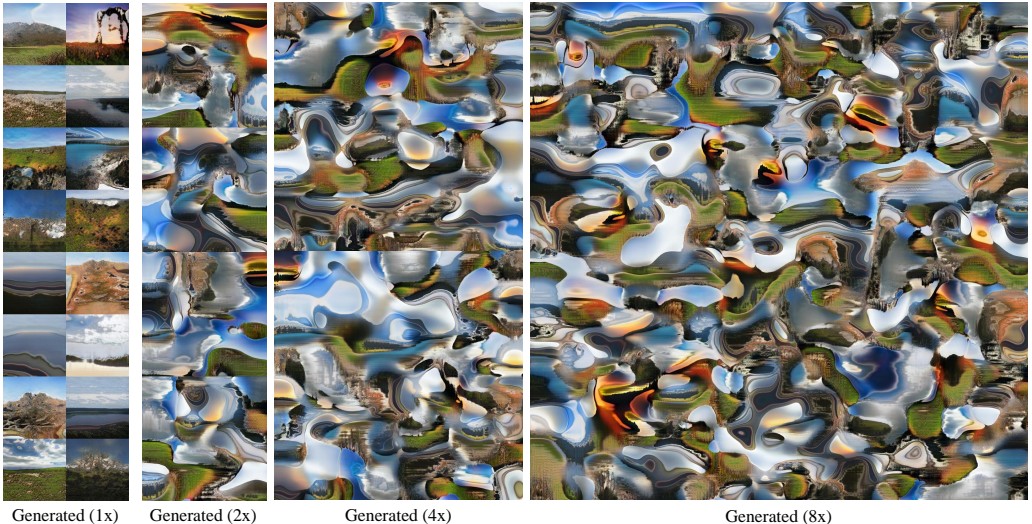

Generated (1x)    Generated (2x)    Generated (4x)    Generated (8x)

Figure 38: **Qualitative results of TileGAN on Flickr-Landscape dataset.** The results show that random texture synthesis models are not directly applicable to real-world image synthesis.

## U    MORE COMPARISONS WITH SINGAN-BASED MODELS

We conduct additional experiments on ConSinGAN (Hinz et al., 2021), a concurrent work that proposes several improvements upon SinGAN. We use the officially released codes and hyperparameters to train the ConSinGAN models. ConSinGAN has two types of training modes, generation and retargeting. The author mentions the "retargeting" mode is more suitable for extending synthesis size in their GitHub release. As shown in Figure 39, similar to SinGAN, ConSinGAN does not have specialized mechanisms to deal with different positional information while tested at a different synthesis size. Therefore, neither of the training modes can produce a plausible global view while tested at extended synthesis sizes.

---

[3]Codes: https://github.com/afruehstueck/tileGAN/blob/0460e228b1109528a0fefc6569b970c2934a649d/networks.py#L274-L294.

[4]Codes: https://github.com/afruehstueck/tileGAN/blob/0460e228b1109528a0fefc6569b970c2934a649d/networks.py#L370-L375.

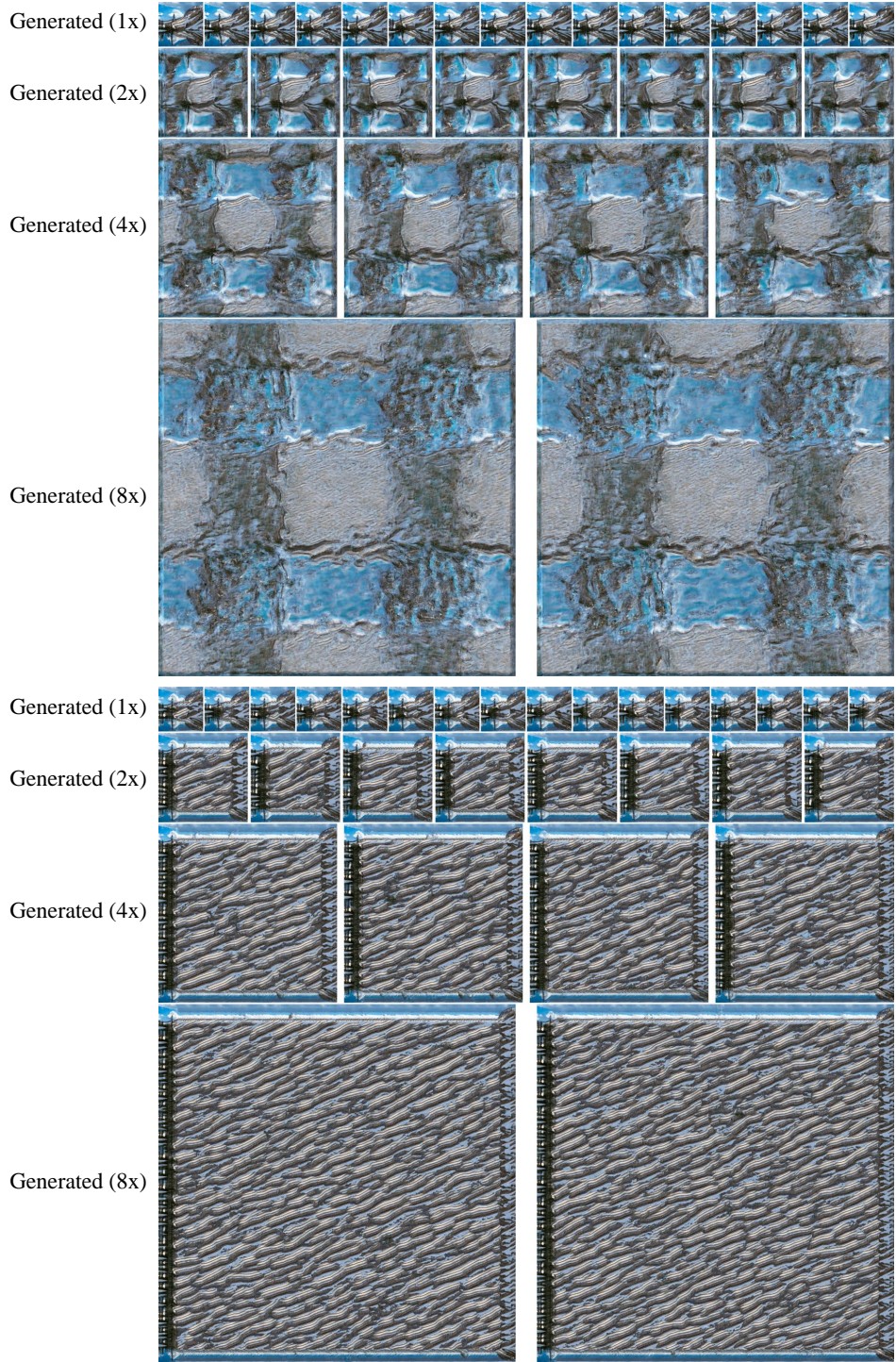

Figure 39: **Qualitative results of ConSinGAN on Flickr-Landscape dataset.** We run ConSinGAN under two different configurations released by the authors, (top) generation and (bottom) retarget. The results show that ConSinGAN inherits similar behaviors from SinGAN and fails to produce images with plausible global structure while synthesizing at larger image sizes.

# V ABLATION: MODE-SEEKING DIVERSITY LOSS ($\mathcal{L}_{\text{div}}$)

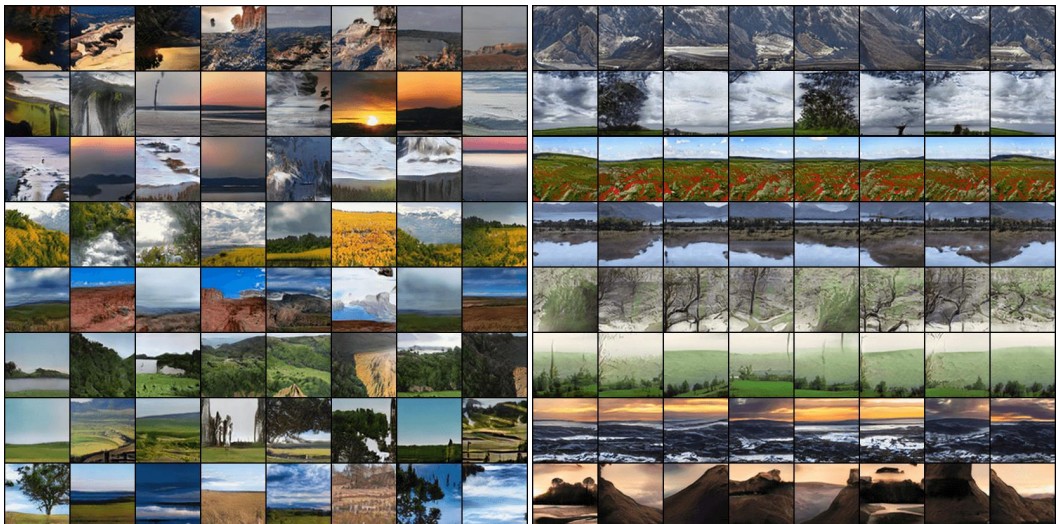

(a) InfinityGAN           (b) InfinityGAN w/o Diversity Loss

Figure 40: **Ablation on the mode-seeking diversity loss.** We show that the mode-seeking diversity loss discourages the model from synthesizing similar appearance at the same coordinate. Both InfinityGAN models are trained on Flickr-Landscape data with 197×197 full-image size and 101×101 patch size, then synthesize at 101×101 pixels at testing. In this figure, all images share the same coordinate grid, and each row shares the same global latent variable. Therefore, only the local latent variables are varying in each row. In Figure (a), regular InfinityGAN shows high diversity in each row, and no obvious structure-coordinate relation is presented. In contrast, in Figure (b), a consistent high-level layout is shared among each row, while the differences between samples are mostly local variations. In particular, the third, sixth, and seventh rows share a similar layout, which is a sign that the model learns a correspondence between the image structure and the coordinates. However, it is difficult to quantify such a problem since the repetition is not an exact repetition of content but a structural/semantical level similarity.

# W ABLATION: AUXILIARY LOSS ($\mathcal{L}_{\text{ar}}$)

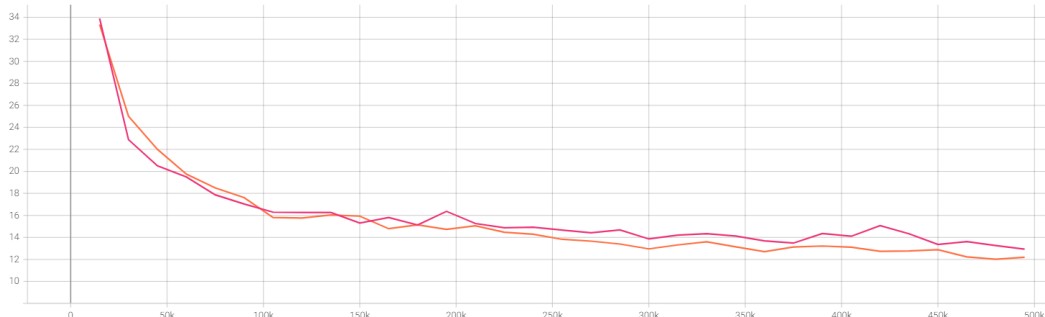

Figure 41: **Ablation on the auxiliary loss.** We show that InfinityGAN with auxiliary loss (orange curve) can provide slight improvement compared to the variant without the auxiliary loss (pink curve). However, such a performance difference is not very significant. We believe the additional supervision in the vertical position should provide important clues in helping the model learn the spatial-varying distribution in the vertical direction. Our approach in modeling such information with two MLP-layers (see Figure 18) may be too naive. Future studies on improving the modeling performance with better loss functions or architecture may improve the overall performance of InfinityGAN further.

