# OpenReview forum: "InfinityGAN: Towards Infinite-Pixel Image Synthesis"
_ICLR.cc/2022/Conference — ICLR 2022 Poster_

### Official Review · Reviewer_1Z86 · 2021-10-31

**Correctness:** 3
**Technical Novelty And Significance:** 3
**Empirical Novelty And Significance:** 3
**Recommendation:** 6
**Confidence:** 4

**Main Review:**

This paper proposes a novel framework, InfinityGAN, for infinitely-wide landscape image generation, combining recent advances on style-based generative adversarial networks and implicit neural representations. Leveraging parallelizable inference, patch-based generative modelling and hand-crafted landscape image priors, the proposed framework can synthesize landscape images of seemingly infinite width and a large height. The paper further proposes a new evaluation metric, ScaleInv FID, and a set of applications for their framework, including image outpainting and inbetweening, and spatial style fusion. The results of InfinityGAN are validated using user studies and some image-space metrics.

The paper proposes very interesting ideas on image generation, including a padding-free StyleGAN architecture, feature unfolding and neural implicit representations for adding variation to the generated image contents, as well as hand-crafted priors on landscape image generation, including positional encodings and disentanglements between global and local styles. The paper is well-written and presented and is extended with a very valuable supplementary material. It includes sufficient details on implementation for allowing reproducibility. The proposed applications could be impactful in the image generation literature.

Overall, this paper shows potential for somewhat significant algorithmic contributions for image generative models, but it falls short on important aspects that need to be adressed before it can be recommended for acceptance. First, several of the claims of the paper should be toned down. The paper argues it proposes a generic framework for infinite image synthesis, but the hand-crafted priors embedded in the method and the datasets used for validation are very specific for landscape photography, making it more limited in its scope than the paper claims. Second, many decisions taken throughout the design of the framework are not well justified or validated empirically, making it hard to understand what the actual contributions of the paper really are.  Finally, this paper misses important references and comparisons to prior work, the comparisons that are present in the paper are somewhat arbitrary and the proposed evaluation metric lacks implementation details and justification.

*Detailed review*

- This paper should tone down its claims of generality. The paper claims to present "a novel framework for arbitrary-sized image generation". However, the design of the method and the datasets used for evaluating it are limited to landscape photography, which shows particularities that InfinityGAN exploits in clever ways but limits its generality. In particular, the hand-crafted prior of self-similarity on the horizontal axis but rapid saturation on the vertical axis only holds for this particular image domain. Therefore, it is unlikely to work in other domains where arbitrary-sized image generation is important, including texture synthesis or satellite photography. This limitation is implicitly acknowledged on the conclusions section, but is not properly discussed throughout the paper.

- Several important references are missing on the related work. In the context of neural implicit representations, relevant recent advances should be included and discussed (eg Fourier Features Let Networks Learn High Frequency Functions in Low Dimensional Domains, Tancik et al, NeurIPS 2020; Implicit Neural Representations with Periodic Activation Functions, Sitzmann et al, NeurIPS 2020).

- Important comparisons with previous work on image generation are missing: TileGAN: Synthesis of Large-Scale Non-Homogeneous
Textures, Frühstück et al, SIGGRAPH 2019; Learning Texture Manifolds with the Periodic Spatial GAN,  Bergmann et al, ICML 2017;  Improved Techniques for Training Single-Image GANs, Hinz et al, WACV 2021. Concurrent work that could be mentioned includes Taming Transformers for High-Resolution Image Synthesis, Esser et al, CVPR 2021. Finally, the paper proposes an inbetweening application, which should be better contextualized on the image inpainting literature.

- Some parts of the design of the framework need to be better empirically demonstrated.
1) What will happen if the self-similarity prior is introduced on both axes?
2) What is the impact of the period T of the positional encoding?
3) What is the impact of the mode-seeking diversity loss introduced to the generator training?
4) Why is the auxiliary task of the generator needed? What happens if it is not done?
5) What will happen when a non-landscape dataset is used?

- With regards to Section 4:
1) The proposed ScaleInv FID seems rather arbitrary and its implementation is not clear. Could the authors elaborate further on this metric? From the current level of detail about this metric, it is quite hard to draw conclusions from it.
2) The comparison with SinGAN is rather unfair to their method. The paper could include more implementation details on how they implement this comparison. It also remains unclear if the improved results of InfinityGAN are due to its inductive biases (hand crafted priors, neural implicit representations, etc), or simply because InfinityGAN is trained on tens of thousands of images, whilst SinGAN is trained on a single image.
3) The user study needs further explanation. What are the demographics of this user set? Which images were they presented with (eg what dataset). What were the questions they were asked?
4) Why is the inception score (IS) used on Table 2 but not on Table 1? Could the authors extend these analyses with other metrics, including SSIM or LPIPS?

In terms of results, on the Supplementary Material, it is clear that InfinityGAN struggles to add variation to the uppermost parts of the images (eg Figures 20 and 21). This is likely due to the tanh() operation done to the vertical coordinates, which limits the variation on this axis compared to the horizontal, positonal-encoded, axis. Could the authors elaborate on this and maybe show more examples on vertical expansion?

I have other minor comments:
- Was any type of data augmentation considered, rather than random cropping? This could alleviate the need for such massive datasets. (Training Generative Adversarial Networks with Limited Data, Karras et al 2020)
- It would be really interesting to see spatial fusion and outpainting applications in the LSUN dataset and other architectural datasets. Did the authors consider this?
- Could this method be extended for 360º photography for VR applications? For example, by making the positional encoding periodic for both axes? It would be great if authors could ellaborate on this.


** Update after rebuttal period**


The authors have successfully adressed many of my concerns during the rebuttal period, including new extensive comparisons with other methods, ablation studies, discussion of limitations and clarifications on the implementation of the Scale-Invariant FID metric. I am overall more positive about this submission and thereby recommend it for acceptance, as this paper proposes novel and well-evaluated ideas, despite some of the claims being somewhat oversold. The responses to concerns by other reviewers were also convincing.



**Summary Of The Paper:**

This paper proposes a novel framework, InfinityGAN, for infinitely-wide landscape image generation, combining recent advances on style-based generative adversarial networks and implicit neural representations. Leveraging parallelizable inference, patch-based generative modelling and hand-crafted landscape image priors, the proposed framework can synthesize landscape images of seemingly infinite width and a large height. The paper further proposes a new evaluation metric, ScaleInv FID, and a set of applications for their framework, including image outpainting and inbetweening, and spatial style fusion. The results of InfinityGAN are validated using user studies and some image-space metrics.

**Summary Of The Review:**

This paper proposes very interesting ideas on image generation, including a padding-free StyleGAN architecture, feature unfolding and neural implicit representations for adding variation to the generated image contents, as well as hand-crafted priors on landscape image generation, including positional encodings and disentanglements between global and local styles. It includes sufficient details on implementation for allowing reproducibility. The proposed applications could be impactful in the image generation literature.

Overall, even though this paper shows potential for somewhat significant algorithmic contributions for image generative models, it falls short on important aspects that need to be adressed before it can be recommended for acceptance. First, several of the claims of the paper should be toned down. The paper argues it proposes a generic framework for infinite image synthesis, but the hand-crafted priors embedded in the method and the datasets used for validation are very specific for landscape photography, making it much more limited in its scope than the paper claims to be. Second, many decisions taken throughout the design of the framework are not well justified or validated empirically, making it hard to understand what the actual contributions of the paper really are.  Finally, this paper misses important references and comparisons to prior work, the comparisons that are present in the paper are somewhat arbitrary and the proposed evaluation metric lacks implementation details and justification.

** Update after rebuttal period**
The authors have adressed many of my concerns during the rebuttal period, including new extensive comparisons with other methods, ablation studies and discussion of limitations. I am overall more positive about this submission and thereby recommend it for acceptance.

---

> ### Author Response · Authors · 2021-11-16
> **Responses to Reviewer 1Z86 (5/5)**
>
> > “Could the authors extend these analyses with other metrics, including SSIM or LPIPS?”
>
> PSNR, SSIM, and LPIPS are metrics assuming an image ground-truth exists and the prediction is expected to be pixel-perfect-aligned with the ground-truth, which is usually not the case under our problem setting.
>
> ---
>
> > "InfinityGAN has lower variation in the uppermost parts of the images. Could the authors elaborate on this and maybe show more examples on vertical expansion?"
>
> The lower variation in the upper part is expected, as the only valid context in the uppermost parts of the image is the sky, the clouds, and the twilight (e.g., the top-left image in Figure 20). It may also be a consequence of some dataset biases since we mainly crawl images from landscape photography groups instead of groups that focus on sky photos.
>
> In the following anonymous link, we provide 400 random sampled sky patches: [https://drive.google.com/drive/folders/1_k6GTdCN1EvtKcfdpnLNmqNdK27B7z4N?usp=sharing]. The images are sampled from the InfinityGAN model trained on Flickr-Landscape with the patch size 101 and full image 197 setting. We synthesize 400 images at 1024x1024 pixels at the same coordinate, then crop the top-left 256x256 region from each of the images. We observe a high diversity in the cloud structure and a satisfactory textural variety.
>
> ---
>
> > "Was any type of data augmentation considered, rather than random cropping? This could alleviate the need for such massive datasets. (Training Generative Adversarial Networks with Limited Data, Karras et al 2020)"
>
> We only additionally apply a random horizontal flipping augmentation in our training pipeline. We have also tried the Adaptive Data Augmentation (ADA) at the early stage of the project, but then it is removed as it did not bring significant improvements. Furthermore, since our model is trained with an adequate amount of data (400,000 images), and we also find that the performance is just slightly better than training with fewer (100,000) images. Therefore, we believe data augmentation may only add on a limited performance improvement.
>
> ---
>
> > "It would be really interesting to see spatial fusion and outpainting applications in the LSUN dataset and other architectural datasets. Did the authors consider this?"
>
> Here we provide some spatial-style-fusion synthesis results on LSUN bridge and tower via the following anonymous link: [https://drive.google.com/drive/folders/1aQS49QF-agi9ngXjm7WkxCDlEkpNNY_s?usp=sharing]. Note that we also provide a failure sample (the last row of each image) for both datasets.
>
> Since LSUN is relatively an object-centric dataset, the global latent variables in InfinityGAN mainly correspond to the appearances of the buildings. Therefore the fusion results are visually similar to switching from a particular style of buildings to another. The failure case of the bridge shows that the model cannot preserve the structural continuity across the fusion area when the building structures on the two sides are impossible to connect, and the tower failure case shows that the generator may fail to interpret the interpolated features if the two styles of buildings are too different.
>
> ---
>
> > "Could this method be extended for 360º photography for VR applications?"
>
> It is possible, but this may be mainly attributed to the coordinate-conditioning model design. In fact, COCO-GAN has presented synthesizing horizontally cyclic panorama synthesis (i.e., cylindrical coordinates). Sharing the similar patch-wise generation and the coordinate-conditioning design with COCO-GAN, InfinityGAN is expected to achieve the described application with a tailored ball-shaped coordinate design and 360º image datasets.

---

> > ### Comment · Reviewer_1Z86 · 2021-11-20
> > **Additional question**
> >
> > I thank the authors for their detailed response and new results. I have an additional question regarding data augmentation. The authors claim that the ADA policy did not add significant improvements. Is this true for every dataset (eg LSUN dataset, satellite image dataset), or only for the landscape dataset that was used throughout the paper, which contains enough images to make data augmentation redundant? It would be valuable if authors could elaborate on this.

---

> > > ### Author Response · Authors · 2021-11-21
> > > **Responses regarding the ADA experiments**
> > >
> > > We only ablate the ADA policy on the Flickr-Landscape dataset. Since most of the datasets used in our paper are with a good amount of samples, therefore at the early stage of the project, we anticipate the augmentation techniques may only provide a limited amount of improvement. Furthermore, the precise performance gain may also require averaging the performance among multiple trials to become accurate. Considering it has a weak connection to our main contributions, we decided not to invest computational resources in pursuing it.

---

> ### Author Response · Authors · 2021-11-16
> **Responses to Reviewer 1Z86 (4/5)**
>
> > “The comparison with SinGAN is rather unfair to their method. The paper could include more implementation details on how they implement this comparison. It also remains unclear if the improved results of InfinityGAN are due to its inductive biases (hand crafted priors, neural implicit representations, etc), or simply because InfinityGAN is trained on tens of thousands of images, whilst SinGAN is trained on a single image.”
>
> First, we want to make corrections to the SinGAN quantitative values as the table below. The wrong FID statistics were cached and affected its correctness (and only SinGAN is affected as it uses a standalone codebase). Nevertheless, the conclusion that SinGAN cannot do well at extended synthesis sizes still stands after resolving the issue.
>
> (*SI-FID = ScaleInv FID)
>
> |                     |  FID Train | SI-FID 1x |  SI-FID 2x |  SI-FID 4x |  SI-FID 8x |
> |---------------------|-------:|-------:|-------:|--------:|--------:|
> | SinGAN (old, buggy) | 22.39 | 22.39 | 73.23 | 150.80 | 214.32 |
> | SinGAN (fixed)      |  4.21 |  4.21 | 57.10 | 145.12 | 210.22 |
> | InfinityGAN         | 11.03 | 21.84 | 28.83 |  61.41 | 121.18 |
> | | | | | | |
>
> We use the officially released SinGAN codes, and use the flags for extending the synthesis size provided by the author: https://github.com/tamarott/SinGAN/blob/286d3cd51cc327381737844d330348ec97577e60/random_samples.py#L16-L17
>
> We particularly compare with SinGAN as the paper covers the concept of synthesizing images at arbitrary sizes, which has a certain level of relation to infinite-pixel image synthesis. We show that SinGAN produces texture-like and unstructured pixels in the image center when the synthesis size is significantly larger than its training size. Therefore it is not applicable to the infinite-pixel image synthesis task. Furthermore, while considering models trained with a large set of images, we conduct experiments on StyleGAN2+NCI in two setups: training with full images and training with patches. In Figure 4 and Figure 19, both SinGAN and StyleGAN2+NCI synthesize images with texture-like repetitive appearances.
>
> ---
>
> > “The user study needs further explanation. What are the demographics of this user set? Which images were they presented with (eg what dataset). What were the questions they were asked?”
>
> We use Two-alternative forced choice (2AFC) between InfinityGAN and other baselines on the Flickr-Landscape dataset. Each participant is asked to answer 30 pairs of queries. For each query, we show two separate grids of 16 random samples from each of the comparing methods. And the participant is asked to select “the one you think is more realistic and overall structurally plausible.”
>
> Unfortunately, we did not record the quantitative demographics of the participants. The participants are researchers and university students with basic knowledge in computer vision, and none of the participants has prior knowledge of our method or problem setup.
>
> ---
>
> > “Why is the inception score (IS) used on Table 2 but not on Table 1?”
>
> Inception Score is traditionally more frequently used in object-centric datasets such as ImageNet or CelebA. We omit the quantity in Table 1 as it is less frequently reported in recent unconditional image synthesis literature. Here we report the IS values in the table below. We can observe that certain quantities are indeed counter-intuitive. Such as StyleGAN2+NCI trained with a smaller patch size result in higher IS, and COCO-GAN yields a high IS despite its visual quality is obviously worse than StyleGAN2+NCI (which is reflected in FID, but not IS).
>
> | Model |  Full-Image Size | Extra resize (Section 4.1)| Full-Image FID (↓) |  Full-Image IS (↑)  |
> |-------------------|--------------:|---------:|-------------:|-------------:|
> | SinGAN            |  128 |   X |  4.21 | 46.38+-1.38 |
> | COCO-GAN          |  128 |   X |   41.32 | 35.95+-0.56 |
> | StyleGAN2+NCI     |  128 |   X |    4.19 | 29.66+-4.19 |
> | StyleGAN2+NCI  (Patched) |  128 |    X |    21.06 | 37.36+-1.03 |
> | StyleGAN2+NCI+PFG |  197 |   V |   90.79 | 14.00+-0.23 |
> | InfinityGAN       |  197 |   V |    21.84 | 26.92+-0.62 |

---

> > ### Comment · Reviewer_1Z86 · 2021-11-20
> > **Reporting this information**
> >
> > I thank the authors for the response. I suggest the authors to update their manuscript with the correct SinGAN scores, and add the new details on the user study.

---

> ### Author Response · Authors · 2021-11-16
> **Responses to Reviewer 1Z86 (3/5)**
>
> > “What is the impact of the mode-seeking diversity loss introduced to the generator training?”
>
> The diversity loss enforces higher structural variations at the same spatial location (i.e., coordinates). This is critical for InfinityGAN as we use a periodic coordinate system in the horizontal direction.
>
> Here we provide some qualitative samples of InfinityGAN with and without diversity loss via the following anonymous link: [https://drive.google.com/drive/folders/17UGJk5sGY-GFh34PcIT5D6qJ86wWhx3h?usp=sharing]. The models are trained with 160k iterations, which is 20% of a full training episode. For both of the images, each row shares the same coordinates and global latent variable, with the randomly sampled local latent variable. This is the scenario that the diversity loss aims to regularize in Eq. (2).
>
> In InfinityGAN-NoDiversiyLoss.png, we can notice that a consistent high-level layout is shared among each row, while the differences between samples are mostly local variations. And in particular, you can notice that the first, second, fifth, and eighth rows are sharing a similar layout, which is a sign that the model learns a correspondence between the image structure and the coordinates. In contrast, InfinityGAN.png shows high diversity in each row, and no obvious structure-coordinate relation is presented. However, it is difficult to quantify such a problem since the repetition is not an exact repetition of content, but a structural/semantical level similarity.
>
> We will add a related discussion in the Appendix when the model is fully trained.
>
> ---
>
> > “Why is the auxiliary task of the generator needed? What happens if it is not done?”
>
> The auxiliary task aims to reinforce the fact that the distribution of content is spatially varying in the vertical axis. In our early-stage pilot tests, the auxiliary task was a crucial component to synthesize plausible results in vertical dirction. We revisit the component and conduct ablation analysis on our latest model. To our surprise, the current InfinityGAN has become robust enough to learn the global appearance without the assistance of the auxiliary task. Here we show a plot of the FID curve through time via the following anonymous link: [https://drive.google.com/drive/folders/1Ro4PNgIHSa8_ExNbTzJ2b29S4chRcwzY?usp=sharing]. The trend of the InfinityGAN-NoAC seems to converge to a similar endpoint to the ablation target, InfinityGAN. We are still assessing the necessity of the auxiliary loss, but its existence does not seem to make a critical impact on the synthesis results.
>
> We will make a related section in the Appendix when the model is fully trained.
>
> ---
>
> > “What will happen when a non-landscape dataset is used?”
>
> In Figure 5, Figure 27, and Figure 28, we additionally show that InfinityGAN can still work on the LSUN bridge and LSUN tower, despite the two datasets being close to object-centric datasets.
> Please also refer to the satellite image dataset that we demonstrate InfinityGAN is applicable to texture-synthesis datasets.
>
> ---
>
> > “The proposed ScaleInv FID seems rather arbitrary and its implementation is not clear. Could the authors elaborate further on this metric?”
>
> Different from common image synthesis tasks where a reference distribution (i.e., real images) is easily accessible, the infinite-pixel image synthesis task asks the model to generalize to arbitrary synthesis sizes. Therefore, it is not feasible to obtain reference distributions for each synthesis size. As an alternative, we assume that images synthesized at a larger spatial size (equivalent to a larger field of view) should share a certain level of scale-invariant visual similarity to the training images. For instance, resizing a larger scale mountain ridge to a smaller scale does not significantly alter its visual plausibility or overall structural layout.
>
> Consider FID evaluates the distance between reference distribution $R$ and synthesized distribution $S$ with $\text{FID}(R, S)$. The ScaleInv FID first bilinearly interpolates the size of synthesized images back to the training image size, then computes the FID metric, written as $\text{ScaleInvFID} = \text{FID}(R, \text{interpolate}(S))$.

---

> > ### Comment · Reviewer_1Z86 · 2021-11-20
> > **Including pseudocode in the supplementary material**
> >
> > I thank the authors for the response. The answers are satisfactory. I would suggest to add pseudocode (or actual python code) to the supplementary material for allowing reproducibility of their ScaleInv FID metric.

---

> ### Author Response · Authors · 2021-11-16
> **Responses to Reviewer 1Z86 (2/5)**
>
> > “Several important references are missing on the related work.”
>
> Thanks for the suggestion, we have added the corresponding citations in our updated paper.
>
> ---
>
> > “Important comparisons with previous work on image generation are missing:”
>
> ***A. Texture synthesis related work***
>
> As discussed in the paper (and also related to the earlier question in synthesizing satellite images), texture synthesis models are not directly applicable to real-world image synthesis. We demonstrate the problem by running TileGAN on our Flickr-Landscape dataset. The samples are provided via the following anonymous link: [https://drive.google.com/drive/folders/1tGVESFHwLZQRBSJ7j6JqiPrA5_Zc2kkQ?usp=sharing]. The results show that the random texture synthesis cannot produce any plausible global structure. It is important to note that the images shown in TileGAN and PS-GAN paper with plausible global structure, such as the satellite map of Jurassic Park and the paintings, are all conditioned on an image that gives the blueprint of the global structure.
>
> We use the officially released TileGAN pipeline to synthesize the results with randomly sampled latent variables. We follow the instructions and train a PGGAN model at 256x256 resolution, then test the model to synthesize at 512x512, 1024x1024, and 2048x2048 pixels.
>
> Note that we discovered that TileGAN alters the PGGAN architecture from [residual-based ToRGB branch](https://github.com/afruehstueck/tileGAN/blob/0460e228b1109528a0fefc6569b970c2934a649d/networks.py#L274-L294) to [single ToRGB projection](https://github.com/afruehstueck/tileGAN/blob/0460e228b1109528a0fefc6569b970c2934a649d/networks.py#L370-L375). Such a modification is not described in the TileGAN paper, but can be found by diagnosing the model checkpoints released by the authors. The modification leads to significant visual-quality degradation. However, even without the visual-quality degradation, the lack of structural clues makes TileGAN impossible to infer a coherent global structure while synthesizing at larger image sizes.
>
> ***B. ConSinGAN***
>
> We use the officially released codes and hyperparameters to train the ConSinGAN models. The results are presented in the following anonymous link: [ https://drive.google.com/drive/folders/1lm0dyHLHuJYmTNVhwgeKxlFPuNsnGAAh?usp=sharing ]. Each figure consists of four rows, representing four different testing scales (1x, 2x, 4x and 8x, corresponding to 250x250, 500x500, 1000x1000, 2000x2000 pixels).
>
> ConSinGAN has two types of training modes, generation and retargeting. The author mentions the “retargeting” mode is more suitable for extending synthesis size in their GitHub release. However, similar to SinGAN, ConSinGAN does not have specialized mechanisms to deal with different positional information while tested at a different synthesis size. Therefore, neither of the training modes can produce a plausible global view while tested at extended synthesis sizes.
>
> ***C. VQGAN and image inpainting***
>
> We have added a citation to VQGAN (Taming Transformers for High-Resolution Image Synthesis) and image inpainting in the related work section.
>
> ---
> ---
>
> > “What will happen if the self-similarity prior is introduced on both axes?”
>
> As the results we show above, the choice of the coordinate system (or the self-similarity prior mentioned here) is a dataset-dependent hyperparameter. We may use a periodic coordinate system in both directions in texture synthesis datasets such as the satellite image we have presented. But using a periodic coordinate system in both directions on the landscape dataset may lead to a periodically appearing horizontal line, which is not physically reasonable nor realistic. Therefore we directly adopt the saturated coordinate system (i.e., tanh) in the vertical direction in our framework.
>
> ---
>
> > “What is the impact of the period T of the positional encoding?”
>
> This is a hyperparameter that we do not tweak or explore. In general, any value that is not too small that leads to repeating coordinates within an image patch, nor too large that results in small differences between consecutive coordinates where numerical errors occur should result in a reasonable performance.

---

> > ### Comment · Reviewer_1Z86 · 2021-11-20
> > **Valuable comparisons**
> >
> > I thank the authors for running these comparisons and updating the related work section. I think the comparisons with TileGAN and ConSinGAN methods is quite valuable for further understanding the potential impact of InfinityGAN in the literature and I suggest the authors to include these results in the supplementary material of their paper.

---

> ### Author Response · Authors · 2021-11-16
> **Responses to Reviewer 1Z86 (1/5)**
>
> We sincerely thank you for the constructive feedback from the reviewer, which raises several important points we would like to further discuss.
>
> ---
>
> > “First, several of the claims of the paper should be toned down. The paper argues it proposes a generic framework for infinite image synthesis, but the hand-crafted priors embedded in the method and the datasets used for validation are very specific for landscape photography, making it more limited in its scope than the paper claims.”
>
> > “Therefore, it is unlikely to work in other domains where arbitrary-sized image generation is important, including texture synthesis or satellite photography.”
>
> The choice of a coordinate system is a hyperparameter in our framework. We add a new paragraph (Appendix C.1) to discuss the choice of coordinates. Currently, we can roughly split all image datasets used in the image synthesis tasks into three categories: object-centric, texture synthesis, and scenery images. As shown in the following table:
>
> | **Dataset Type**        | `Object-centric` | `Texture`            | `Scenery`          |
> |-|-|-|-|
> | **Example**              | ImageNet, CelebA     | Texture, Satellite                    | Landscape             |
> | **Spatial Distribution of Content** | N/A            | Spatially agnostic | Spatially varying in vertical direction |
> | **Horizontal  Coordinate System**          | Constant       | Periodic (e.g., sin/cos)    | Periodic (e.g., sin/cos) |
> | **Vertical  Coordinate System**            | Constant       | Periodic (e.g., sin/cos)           | Saturate (e.g., tanh) |
> |  |  |  |  |
>
> Note that the LSUN bridge and tower datasets we show in the paper are of the grey area between scenery and object-centric. The concept of the scene itself should be classified as scenery, but the two datasets always assume an object is explicitly around the center of the image, which is the dataset bias of an object-centric dataset.
>
> Although our framework can be applied to all three categories, the expected outcome of applying InfinityGAN on object-centric datasets such as CelebA is ill-defined. On the other hand, we here show the results of applying InfinityGAN on the satellite image dataset (from Pix2Pix paper, consisting of 2194 images). The real images and synthesis results can be found in the following anonymous link: [https://drive.google.com/drive/folders/1XyVfvIHVKCobDV-PIap5LMQSbnY9mNcg?usp=sharing]. We run the experiment in two setups: (a) an intuitive setup with periodic coordinates in both directions (trained 207k iterations), and (b) using a saturated coordinate system (tanh) in the vertical direction and a periodic coordinate system in the horizontal direction (trained 90k iterations). The results show that InfinityGAN works in both setups.
>
> Although InfinityGAN can work on texture synthesis tasks, we do not want to claim or remark it as a contribution as many previous methods in texture synthesis already excel in the task.
>
> ---
>
> > “Second, many decisions taken throughout the design of the framework are not well justified or validated empirically, making it hard to understand what the actual contributions of the paper really are.”
>
> We believe that one of our major contributions is proposing the task of learning to synthesize real-world images at arbitrary sizes while the model is trained at a fixed and finite field of view. As discussed in the related work section, the most similar task in the existing literature is texture synthesis. But texture synthesis methods are not directly applicable to real-world images. Therefore, we propose a structure synthesis model that renders the global structure before texture filling with our texture synthesizer. We believe that our implementation is only one of the possible solutions to the problem. Accordingly,  we majorly focus our ablation on the importance of correcting the positional information, which is the major issue that most of the methods will come across toward the infinite-pixel image synthesis problem.
>
> Note that we further conduct two ablations (removing auxiliary loss, removing mode-seeking loss) as requested in other questions below.

---

> > ### Comment · Reviewer_1Z86 · 2021-11-20
> > **Suggestion to add this comparison to the supplementary material**
> >
> > I thank the authors for the response and the efforts in presenting this discussion and results on satellite images. I think they provide interesting insights on the capabilities of their proposed method. I suggest the authors to add both to the supplementary material.

---

> ### Author Response · Authors · 2021-11-21
> **Updated paper with additional supplementary materials**
>
> We have updated the paper with the suggested sections (discussions on satellite images, comparison with TileGAN and ConSinGAN, pseudo-codes for ScaleInv FID, updated SinGAN quantitative values, and more details on the user study) and annotated the modified/added sections with red fonts. Sincerely thanks for all the valuable suggestions.

---

> ### Author Response · Authors · 2021-11-22
> **Updated paper with additional ablation studies**
>
> We have also included the ablation study on the diversity loss and auxiliary loss in Appendix V and W. Sincerely thanks for your suggestions and patience.

---

> ### Comment · Reviewer_1Z86 · 2021-12-03
> **Response to authors**
>
> I thank the authors for their exhaustive response to my questions. I think the new additions to the manuscript improve its scope and quality, I am now more positive about this manuscript and have updated my review and rating accordingly.

---

### Official Review · Reviewer_TkQk · 2021-11-01

**Correctness:** 4
**Technical Novelty And Significance:** 3
**Empirical Novelty And Significance:** 3
**Recommendation:** 8
**Confidence:** 4

**Details Of Ethics Concerns:**

No specific concerns with this work, but it is important to flag that approaches to generate images can be always misused in certain context. Perhaps the authors could briefly mention this.

**Main Review:**

+ The paper advances the state of art in image synthesis and there are multiple contributions that could be re-used in other contexts. For example, I really liked the intuition behind the padding-free generator and how the authors showed its effectiveness in both quantitative and qualitative results.
+ The paper is well written and justified. All the technical details are well elaborated and additional supplementary material provides all the details needed to ensure reproducibility. The contributions seem to be well placed with respect to the state of art
+ The experimental section is impressive, plenty of examples showing the effectiveness of the approach. The supplementary materials show multiple comparisons and visuals as well as implementation details for the spatial style fusion.

There are a couple, mostly minor weak points, that I hope could help the authors to make the paper even stronger:
- Evaluation on temporal consistency: it would be great to assess how the method performs on temporal sequences and assess how stable/consistent it is. I understand that the method was not designed for this purpose, but could still provide a baseline for future research.
- Ablation study: there are lots of interesting contributions here, from architecture specific modules to loss functions. I would suggest to elaborate more the ablation study section, which mostly focuses on the padding-free generator at the moment.

**Summary Of The Paper:**

The paper proposes a framework to generate high resolution images of outdoor scenes with any desired style. The approach extends image synthesis techniques and proposes a system that uses both global and local information. This ensures that the model can be trained using low resolution patches (e.g. 101 x 101) and, at inference time, it can be used to generate high resolution renderings: images up to 4096 x 4096 are showed.

The combination of the global and local appearance, coupled with interesting modules such as the padding-free generator, ensure high quality results that are consistently outperforming other approaches.

**Summary Of The Review:**

This paper is a solid submission, that really pushes the state of art in image synthesis for outdoor scenes. The problem is well justified and placed with respect to related work and the contributions are well described. The results prove that the method is outperforming baselines with comparisons on both quantitative and qualitative results. Given the complexity of the framework, I would have appreciated a more in-depth analysis (i.e. ablation study) of the various components to help other researchers to reproduce the results and focus on the right modules, however I acknowledge that authors did their best with experiments and evaluations: the supplementary material is actually impressive.

I would suggest authors to also show some results on temporal sequences, to assess temporal consistency across frames and/or long sequences. This would also help to establish a baseline for future work in this area.

---

> ### Author Response · Authors · 2021-11-16
> **Responses to Reviewer TkQk**
>
> We sincerely thank you for the positive and constructive feedback. We have added the following ethics statement to our updated paper:
>
> ```
> Our work follows the General Ethical Principles listed at ICLR Code of Ethics (https://iclr.cc/public/CodeOfEthics).
>
> The research in generative modeling is frequently accompanied by concerns about the misuse of manipulating or hallucinating information for improper use. Despite none of the proposed techniques aiming at improving manipulation of fine-grained image detail or hallucinating human activities, we cannot rule out the potential of misusing the framework to recreate fake scenery images for any inappropriate application. However, as we do not drastically alter the plausibility of synthesis results in the high-frequency domain, our research is still covered by continuing research in ethical generative modeling and image forensics.
> ```
>
> ---
>
> > “Evaluation on temporal consistency: it would be great to assess how the method performs on temporal sequences and assess how stable/consistent it is. I understand that the method was not designed for this purpose, but could still provide a baseline for future research.”
>
> The temporal consistency is fundamentally different from the pixel consistency we tackle in this paper. The root cause of pixel inconsistency from baselines we have shown in the paper is the ill-posed positional information from the zero-paddings. On the other hand, the temporal model in video synthesis is usually a sequential model (e.g., LSTM or Transformer) that does not rely on padding-based positional information similar to CNN models. Therefore adopting our framework in video synthesis is unlikely to provide improvement in temporal consistency.
>
> Meanwhile, in terms of adopting neural implicit representation in the temporal domain, we believe it is not a straightforward adaptation. We found a related work, DIGAN (https://openreview.net/forum?id=Czsdv-S4-w9), submitted to ICLR’22. It can be an interesting future research direction in combining InfinityGAN and DIGAN.
>
> ---
>
> > “Ablation study: there are lots of interesting contributions here, from architecture specific modules to loss functions. I would suggest to elaborate more the ablation study section, which mostly focuses on the padding-free generator at the moment.”
>
> We want to start by highlighting that we have included an ablation on feature unfolding in Appendix Q. Here we provide an additional qualitative ablation to the diversity loss.
>
> The diversity loss enforces higher structural variations at the same spatial location (i.e., coordinates). This is critical for InfinityGAN as we use a periodic coordinate system in the horizontal direction.
>
> Here we provide some qualitative samples of InfinityGAN and InfinityGAN without diversity loss via the following anonymous link: [https://drive.google.com/drive/folders/17UGJk5sGY-GFh34PcIT5D6qJ86wWhx3h?usp=sharing]. The models are trained with 160k iterations, which is 20% of a full training episode. For both of the images, each row shares the same coordinates and global latent variable, only the local latent variable is randomly sampled. This is exactly the scenario that the diversity loss aims to regularize in Eq. (2).
>
> In InfinityGAN-NoDiversiyLoss.png, we can notice that a consistent high-level layout is shared among each row, while the differences between samples are mostly local variations. And in particular, you can notice that the first, second, fifth, and eighth rows are sharing a similar layout, which is a sign that the model learns a correspondence between the image structure and the coordinates. In contrast, InfinityGAN.png shows high diversity in each row, and no obvious structure-coordinate relation is presented. However, it is difficult to quantify such a problem since the repetition is not an exact repetition of content, but a structural/semantical level similarity.
>
> We will add a related discussion in the Appendix when the model is fully trained.

---

> > ### Comment · Reviewer_TkQk · 2021-11-30
> > **Response to authors**
> >
> > Thank you for your response and the additional details provided, I have really appreciated the authors' effort here. I am very happy with the paper and in support of acceptance.

---

### Official Review · Reviewer_gucW · 2021-11-02

**Correctness:** 4
**Technical Novelty And Significance:** 3
**Empirical Novelty And Significance:** 3
**Recommendation:** 8
**Confidence:** 3

**Main Review:**

I think this is a very solid paper. The writing is good and the proposed model seems novel.  The task is very interesting and the authors addressed several challenges in very smart ways. For example, I think generating two patches that can be combined seamlessly is hard. The authors proposed a modification to StyleGAN to eliminate zero padding. Position information is further added to generate plausible results.
The experiments are sufficient and comprehensive. A lot of details are provided in the supplementary material. Applications including spatial style fusion, multimodal outpainting, and image inbetweening are also impressive.


The major concern I have is about the problem formulation. The whole image generation pipeline involves 3 random variables as input: $z_g$, $z_l$, and $z_n$. $z_n$ is used as noise input in StyleGAN. The structural latent variable $z_S$ is generated from $G_S(z_g, z_l, c)$. However, $z_S$ is only used in $G_T$, while there is another input $z_g$ in G_T. I was wondering if the $z_g$ input in $G_S$ is necessary? As shown by Fig 6, $z_l$ determines the layout while $z_g$ in $G_T$ for style, I did not see the role of $z_g$ in $G_S$. In other words, what if we use different $z_g$ in $G_S$ and $G_T$? Which $z_g$ will determine the style?

Here are some minor issues that will not affect my rating on this paper. But I would appreciate it if the authors could share more insight:

1 In the previous work section, I wondered if the authors also want to include autoregressive models like PixelCNN. These models can generate images of arbitrary sizes or shapes by nature. The authors did cite the VQVAE2 paper and mentioned it could only generate images of fixed sizes.

2 I found the visual result on the Flickr-Landscape data is better than that on the LSUN bridge or tower datasets. For example, Fig 27 top left shows repetitive tower patterns. Is this limited by the proposed pipeline, or the datasets, or StyleGAN?

**Summary Of The Paper:**

This paper proposed a new model called InfinityGAN, for generating images of arbitrary sizes. These images are generated holistically: the authors proposed using a structure synthesizer first to generate a latent structural representation. Then a modified StyleGAN based texture synthesizer generates the image patches at the corresponding locations. The positional encoding and PFG enable generating patches in an arbitrary large image and aggregating these patches seamlessly. The experiments on the Flickr-Landscape dataset show promising large image synthesis performance. On the Place365 and Flickr-Scenery datasets, the experiment results show superior image outpainting performance. The authors also demonstrated other applications including spatial style fusion and image inbetweening.

**Summary Of The Review:**

I did not find big issues with this submission. It is a very good paper, and I think the proposed techniques can benefit other large image generation research. I am only concerned about the $z_g$ in both $G_S$ and $G_T$, which I think might be redundant. Therefore I recommend accepting this submission.

---

> ### Author Response · Authors · 2021-11-16
> **Responses to Reviewer gucW**
>
> We sincerely thank you for the positive and constructive feedback. We have added the suggested citations to our paper. Here we respond to the raised concerns:
>
> ---
>
> > “Missing citation to PixelCNN.”
>
> We add the following citations and discussion in our updated paper:
> “Finally, autoregressive models (cite PixelCNN, VQVAE2, VQGAN) can theoretically synthesize at arbitrary image sizes. Despite VQVAE2 and VQGAN showing unconditional images synthesis at 1024$\times$1024 resolution, their application in infinite-pixel image synthesis has not yet been well-explored.”
>
> In fact, both VQVAE2 and VQGAN adopt a CNN-decoder after their PixelCNN synthesis pipeline. The CNN-decoder, with zero-paddings and its positional information, forbids these frameworks from synthesis by parts. Therefore, bounded by the memory constrain, these methods can only synthesize images up to a certain size.
>
> ---
>
> > “I was wondering if the $\mathbf{z}_\mathrm{g}$ input in $G_\mathrm{S}$ is necessary? As shown by Fig 6, $\mathbf{z}_\mathrm{l}$ determines the layout while $\mathbf{z}_\mathrm{g}$ in $G_\mathrm{T}$ for style, I did not see the role of $\mathbf{z}_\mathrm{g}$ in $G_\mathrm{T}$. In other words, what if we use different $\mathbf{z}_\mathrm{g}$ in $G_\mathrm{S}$ and $G_\mathrm{T}$? Which $\mathbf{z}_\mathrm{g}$ will determine the style?”
>
> In fact, the layout is jointly determined with both $\mathbf{z}_\mathrm{l}$ and $\mathbf{z}_\mathrm{g}$ in $G_\mathrm{S}$. A critical difference between $\mathbf{z}_\mathrm{l}$ and $\mathbf{z}_\mathrm{g}$ is that $\mathbf{z}_\mathrm{l}$ affects regions is constrained by the receptive field of $G_\mathrm{S}$, while $\mathbf{z}_\mathrm{g}$ affects the whole image. Furthermore, as $\mathbf{z}_\mathrm{g}$ is injected via feature modulation, it alters the overall distribution of structural features (i.e., $\mathbf{z}_\mathrm{S}$) determined by $\mathbf{z}_\mathrm{g}$. Therefore, it is expected to affect the global-level visual attributes.
>
> We provide a grid of images similar to Figure 6 and Figure 29 in the following anonymous link: [https://drive.google.com/drive/folders/1Mu742ys0LSfAi9kygiut3uQGq4QxXKMY?usp=sharing]. In this figure, the vertical axis is variating global latent in the structure synthesizer, while the horizontal axis is different styles determined by a separate global latent. Noticeably, the global latent controls the global and semantic-level attributes of the image, such as the amount of cloud, the size (can also interpret as distance) to the objects in the scene, and the rockness of the mountain. Note that a certain level of performance degradation or artifacts may be expected in this figure, as the inputs are different from the model training distribution.
>
> ---
>
> > "I found the visual result on the Flickr-Landscape data is better than that on the LSUN bridge or tower datasets. For example, Fig 27 top left shows repetitive tower patterns. Is this limited by the proposed pipeline, or the datasets, or StyleGAN?"
>
> It can be a combination of a dataset bias and a limitation of our framework, which is a part of our future work directions. We want to start with highlighting that this type of repetition does not stick to certain coordinates (i.e. such repetition does not appear at the same coordinate/location among different images) and it spans a very short period in that image sample (not the whole image repeating the same content).
>
> First,  regarding the dataset bias, LSUN bridge and tower are both object-centric datasets. So it is not surprising that the model (a) learns to tight global latent to the identity of the object, and (b) tries to create objects periodically, since the patch-based model trains on real-image patches that almost always contain a part of an object.
>
> Second, it can be a limitation of the discriminator. While our global latent variable tights to an infinite-pixel image, the discriminator still has a finite receptive field in each inference. Therefore the adversarial loss cannot forbid the generator from synthesizing similar or even repetitive context outside its momentary receptive field. This is a fundamental issue for any discriminative model or loss function with an explicit and finite receptive field. We may slightly ease the phenomenon by increasing the discriminator receptive field, but there can always be repetition beyond the designated receptive field.

---

> > ### Comment · Reviewer_gucW · 2021-11-30
> > **Responses to the authors**
> >
> > I deeply appreciate the authors' feedback and the responses are generally satisfactory. Especially I thank the authors for the extra experiments with two different global latent variables and most of my concerns have been addressed.
> >
> > From the extra qualitative results, it is clear the global style is jointly determined by "two" global latent variables in both $G_S$ and $G_T$, though in the current implementation, the authors chose to use the same $z_g$. I also believe investigating the interaction between the same or different $z_g$ in the equation is beyond the scope of this paper.
> >
> > The visualization results have limitations, but they won't hurt the significance of this paper. I am looking forward to future work on this topic.
> >
> > To sum up, I would like to keep my recommendation for acceptance and I am pretty positive about this paper.

---

### Official Review · Reviewer_bTWK · 2021-11-02

**Correctness:** 4
**Technical Novelty And Significance:** 3
**Empirical Novelty And Significance:** 3
**Recommendation:** 8
**Confidence:** 4

**Main Review:**

Strength:
1) The idea to generate infinity-pixel images in a seamless patch-by-patch manner is quite novel.
2) This approach considers both global and local consistency in generating large images. Though straightforward, the designed framework based on it is effective and elegant.
3) The experiments and the appendix are comprehensive to reproduce this work.

Weakness:
No obvious weakness but there are some issues.
1) It is better to add the symbols in Sec 3.2 & 3.3 to Figure 2, e.g., $G_s$, $Z_s$, $G_T$, $p_c$.
2) Although the concurrent work ALIS (SKOROKHODOV ET AL., 2021) has been discussed in appendix A, it is suggested to make a fair comparison in Table 1 with the same metrics in the future version, since ALIS address nearly the same task to generate the infinity-pixel images.


**Summary Of The Paper:**

This paper proposes a novel framework, InfinityGAN, which could generate arbitrary-sized images with infinite pixels.

By disentangling global appearance, local structures and textures, InfinityGAN generates local and global consistent large images with a seamless patch-by-patch paradigm.

Experiments validate its superiority compared to some other baselines, and several applications are raised based on their approaches.

**Summary Of The Review:**

Based on its strength and weakness, I think it is a good and quite novel paper considering both its framework and applications. I recommend accepting this paper.

---

> ### Author Response · Authors · 2021-11-16
> **Responses to Reviewer bTWK**
>
> We sincerely thank the positive and constructive feedback. We have updated Figure 2 in the latest paper version according to the suggestion. However, the location of $\mathbf{z}_\mathrm{S}$ is difficult to label in the figure, so it remains omitted.
>
> ---
>
> > “It is suggested to make a fair comparison with ALIS in Table 1 with the same metrics in the future version.”
>
> Despite InfinityGAN and ALIS sharing a similar concept, it is difficult to make an entirely fair comparison between the two methods. First, ALIS cannot inbetween image patches in the vertical direction for landscape images. For instance, considering landscape or LSUN bridge/tower datasets, a majority of images synthesized with ALIS should contain the horizon line. Vertically stacking these images among each other leads to incorrect image layout, and no inbetweening prediction can resolve such a wrong layout. Second, our ScaleInv FID requires square-shaped synthesis results that extend in all directions and cannot evaluate ALIS results where long-strip-shaped images are generated.
>
> As a fallback solution, we report the $\infty$-FID proposed by ALIS in the fourth paragraph of Appendix A. However, the $\infty$-FID is specifically designed for the patch-inbetweening setting where the two images on two sides do not share the same global context. InfinityGAN assumes a universal global context is shared within each infinite-pixel image, and thus it is not applicable to adopt the $\infty$-FID evaluation in the random synthesis setting. We accordingly only report the InfinityGAN results with the spatial-style-fusion setting. As discussed in the same paragraph, ALIS and our spatial-fusion mechanism form a tradeoff between metric-wise performance and flexibility.

---

> > ### Comment · Reviewer_bTWK · 2021-11-29
> > **Responses to the Author**
> >
> > Thanks for your response and accept suggestions. After reading the feedback, I have no further questions and I still maintain my original rating for this paper.

---

### Decision · Program_Chairs · 2022-01-20

**Decision:**

Accept (Poster)

**Comment:**

3 reviewers recommend accept, 1 rates the paper marginally above acceptance. The authors provided satisfactory answers to criticism -- all in all this is a paper worth accepting at ICLR. Please make sure that criticism in the reviews is adequately addressed in the final version, e.g. include various experimental results in the rebuttal, add the symbols in sec 3.2 & 3.3 to fig. 2, add a related discussion on ablations when the model is fully trained, etc.